# Latent Field Discovery In Interacting Dynamical Systems With Neural Fields

**Miltiadis Kofinas**
University of Amsterdam
m.kofinas@uva.nl

**Erik J. Bekkers**
University of Amsterdam
e.j.bekkers@uva.nl

**Naveen Shankar Nagaraja**
BMW Group
Naveen-Shankar.Nagaraja@bmw.de

**Efstratios Gavves**
University of Amsterdam
egavves@uva.nl

## Abstract

Systems of interacting objects often evolve under the influence of field effects that govern their dynamics, yet previous works have abstracted away from such effects, and assume that systems evolve in a vacuum. In this work, we focus on discovering these fields, and infer them from the observed dynamics alone, *without* directly observing them. We theorize the presence of latent force fields, and propose neural fields to learn them. Since the observed dynamics constitute the net effect of local object interactions and global field effects, recently popularized equivariant networks are inapplicable, as they fail to capture global information. To address this, we propose to *disentangle* local object interactions –which are $\mathrm{SE}(n)$ equivariant and depend on relative states– from external global field effects –which depend on absolute states. We model interactions with equivariant graph networks, and combine them with neural fields in a novel graph network that integrates field forces. Our experiments show that we can accurately discover the underlying fields in charged particles settings, traffic scenes, and gravitational n-body problems, and effectively use them to learn the system and forecast future trajectories.

## 1 Introduction

Systems of interacting objects are omnipresent in nature, with examples ranging from the sub-atomic to the astronomical scale –including colliding particles and n-body systems of celestial objects– as well as settings that involve human activities, governed by social dynamics, like traffic scenes. The majority of these systems does not evolve in a vacuum; instead, systems evolve under the influences of underlying fields. For example, electromagnetic fields may govern the dynamics of charged particles, while galaxies

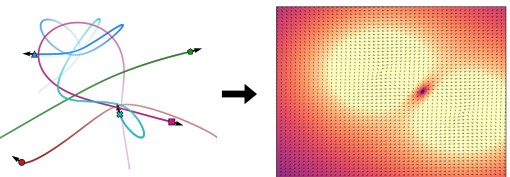

Figure 1: N-body system with underlying gravitational field. We uncover fields that underlie interacting systems using only the observed trajectories.

swirl around supermassive black holes that create gravitational fields. In traffic scenes, the road network and traffic rules govern the actions of traffic scene participants. Despite the ubiquity of fields, previous works on modelling interacting systems have only focused on the *in vitro* case of systems evolving in a vacuum.

Earlier work on learning interacting systems proposed graph networks [3, 23, 40]. Recently, state-of-the-art methods for interacting systems propose *equivariant* graph networks [50, 41, 24, 5, 10] to

37th Conference on Neural Information Processing Systems (NeurIPS 2023).

model dynamics while respecting the symmetries that often underlie them. These networks exhibit increased robustness and performance, while maintaining parameter efficiency due to weight sharing. They are, however, not compatible with underlying field effects, since they can only capture local states, such as relative positions, while fields depend on absolute states (*e.g.* positions or orientations). In other words, *global fields violate the strict equivariance hypothesis*.

Within the context of modelling interacting systems, a function $f$ that predicts future trajectories is SE(3) equivariant –equivariant to the special Euclidean group of translations and rotations– if $f(\mathbf{R}\mathbf{x}+\boldsymbol{\tau}) = \mathbf{R}f(\mathbf{x})+\boldsymbol{\tau}$ for a translation vector $\boldsymbol{\tau}$ and a rotation matrix $\mathbf{R}$. While strict equivariance holds in idealized settings, it does not hold in many real-world settings. That is, even if the symmetries exist in a particular setting, they only manifest themselves in local interactions, yet they are entangled with global effects that stem from absolute states. N-body systems from physics, for example, exhibit E(3) symmetries, since gravitational forces only depend on relative positions. Dynamics, however, may be influenced by external force fields, *e.g.* black holes, which are either unknown or not subject to transformations. Thus, strict equivariance is violated, since equivariant object interactions are *entangled* with global field effects.

We make the following contributions. First, we introduce neural fields to discover global latent force fields in interacting dynamical systems, and infer them by observing the dynamics alone. Second, we introduce the notion of *entangled equivariance* that intertwines global and local effects, and propose a novel architecture that disentangles equivariant local object interactions from global field effects. Third, we propose an approximately equivariant graph network that extends equivariant graph networks by using a mixture of global and local information. Finally, we conduct experiments on a number of field settings, including real-world traffic scenes, and extending state-of-the-art setups from the literature. We observe that explicitly modelling fields is mandatory for effective future forecasting, while their unsupervised discovery opens a window for model explainability.

We term our method *Aether*, inspired by the postulated medium that permeates all throughout space and allows for the propagation of light.

## 2  Background

**Interacting dynamical systems**  An interacting dynamical system comprises trajectories of $N$ objects in $d$ dimensions, $d \in \{2, 3\}$, recorded for $T$ timesteps. The snapshot of the $i$-th object at timestep $t$ describes the state $\mathbf{x}_i^t = [\mathbf{p}_i^t, \mathbf{u}_i^t], i \in \{1, \ldots, N\}, t \in \{1, \ldots, T\}$, where $\mathbf{p} \in \mathbb{R}^d$ denotes the position and $\mathbf{u} \in \mathbb{R}^d$ denotes the velocity, using $[\cdot, \cdot]$ to denote vector concatenation along the feature dimension. We are interested in forecasting future trajectories, *i.e.* predict the future states for all objects and for a number of timesteps. Interacting dynamical systems can be naturally formalized as spatio-temporal geometric graphs [3, 23, 14], $\mathcal{G} = \{\mathcal{G}^t\}_{t=1}^T$, with graph snapshots $\mathcal{G}^t = (\mathcal{V}^t, \mathcal{E}^t)$ at different time steps. The set of graph nodes $\mathcal{V}^t = \{v_1^t, \ldots, v_N^t\}$ describes the objects in the system; $v_i^t$ corresponds to $\mathbf{x}_i^t$. The set of edges $\mathcal{E}^t \subseteq \{(v_j^t, v_i^t) \mid (v_j^t, v_i^t) \in \mathcal{V}^t \times \mathcal{V}^t\}$ describes pair-wise object interactions; $(v_j^t, v_i^t)$ corresponds to an interaction from node $j$ to node $i$. Finally, $\mathcal{N}(i)$ denotes the neighbors of node $v_i$.

**Local coordinate frame graph networks**  Local coordinate frame graph networks have been popularized in recent years [24, 26, 10, 52, 20, 30] as a method to achieve SE(3) –or E(3)– equivariance, due to their low computational overhead and high performance. Kofinas et al. [24] proposed LoCS and introduced local coordinate frames for all node-objects at all timesteps. They define augmented node states $\mathbf{v}_i^t = [\mathbf{p}_i^t, \boldsymbol{\omega}_i^t, \mathbf{u}_i^t]$, where $\boldsymbol{\omega}_i^t$ denotes the angular position of node $i$ at timestep $t$. Kofinas et al. [24] use velocities as a proxy to angular positions, while Luo et al. [26] use another network that predicts latent orientations. Each local coordinate frame is translated to match the target object's position and rotated to match its orientation. Considering the representation of node $j$ in the local coordinate frame of node $i$, denoted as $\mathbf{v}_{j|i}^t$, they first compute the relative positions $\mathbf{r}_{j,i}^t = \mathbf{p}_j^t - \mathbf{p}_i^t$ and then they rotate the state using the matrix representation of the angular position $\mathbf{Q}(\boldsymbol{\omega}_i^t)$:

$$\mathbf{v}_{j|i}^t = \tilde{\mathbf{R}}\big(\boldsymbol{\omega}_i^t\big)^\top \big[\mathbf{r}_{j,i}^t, \boldsymbol{\omega}_j^t, \mathbf{u}_j^t\big], \tag{1}$$

where $\tilde{\mathbf{R}}(\boldsymbol{\omega}_i^t) = \mathbf{Q}(\boldsymbol{\omega}_i^t) \oplus \mathbf{Q}(\boldsymbol{\omega}_i^t) \oplus \mathbf{Q}(\boldsymbol{\omega}_i^t)$, and $\oplus$ denotes a direct sum. LoCS then proposes a graph neural network [42, 25, 13] that uses local states:

$$\mathbf{h}_{j,i}^t = f_e\left(\left[\mathbf{v}_{j|i}^t, \mathbf{v}_{i|i}^t\right]\right), \tag{2}$$

$$\boldsymbol{\Delta}\mathbf{x}_{i|i}^{t+1} = f_v\left(g_v\left(\mathbf{v}_{i|i}^t\right) + C\sum_{j \in \mathcal{N}(i)} \mathbf{h}_{j,i}^t\right), \tag{3}$$

where $f_v$, $f_e$, and $g_v$ are MLPs, and $C = {}^1/{|\mathcal{N}(i)|}$. The output of this graph network comprises differences in positions and velocities from the previous time step, in the local frame of each object. Since these outputs are invariant, LoCS performs an inverse transformation to convert them back to the global coordinate frame and achieve equivariance, $\mathbf{x}_i^{t+1} = \mathbf{x}_i^t + \mathbf{R}(\boldsymbol{\omega}_i^t) \cdot \boldsymbol{\Delta}\mathbf{x}_{i|i}^{t+1}$, where $\mathbf{R}(\boldsymbol{\omega}_i^t) = \mathbf{Q}(\boldsymbol{\omega}_i^t) \oplus \mathbf{Q}(\boldsymbol{\omega}_i^t)$.

**Neural fields** Finally, we make a brief introduction to neural fields. Neural fields, or coordinate-based MLPs, are a class of neural networks that parameterize fields using neural networks (see Xie et al. [54] for a survey). They take as input states like spatial coordinates and predict some quantity. Neural fields can learn prior behaviors and generalize to new fields via conditioning on a latent variable $\mathbf{z}$ that encodes the properties of a field. Perez et al. [33] proposed Feature-wise Linear Modulation (FiLM), a conditioning mechanism that modulates a signal. It comprises two sub-networks $\alpha, \beta$ that perform multiplicative and additive modulation to the input signal, and can be described by $\text{FiLM}(\mathbf{h}, \mathbf{z}) = \alpha(\mathbf{z}) \odot \mathbf{h} + \beta(\mathbf{z})$, where $\mathbf{z}$ is the conditioning variable, $\mathbf{h}$ is the signal to be modulated, and $\alpha, \beta$ are MLPs that scale the signal, and add a bias term, respectively.

## 3 Method

In this section, we present our method, termed *Aether*. First, we describe the notion of entangled equivariance, and introduce our architecture that disentangles global field effects from local object interactions. Then, we continue with the description of the neural field that infers latent fields by observing the dynamics alone. Finally, we formulate approximately equivariant global-local coordinate frame graph networks. We note that throughout this work, we focus on fields that are unaffected by the observable objects and their interactions thereof.

### 3.1 Aether

Interacting dynamical systems rarely evolve in a vacuum, rather they evolve under the influence of external field effects. While object interactions depend on local information, the underlying fields depend on global states. On the one hand, locality in object interactions stems from the fact that dynamics obey a number of symmetries. By extension, object interactions are equivariant to a particular group of transformations. On the other hand, field effects are non-local; they depend on absolute object states. Thus, strict equivariance is violated, since equivariant object interactions are entangled with global field effects. We refer to this phenomenon as *entangled equivariance*.

As an example, in Figure 2 we observe a system of two objects that evolve in a gravitational field. The arrows positioned *on* the objects represent the forces exerted on them. One constituent of the net force is caused by object interactions, and is thus equivariant, while the other can be attributed to the gravitational pull. However, we can only observe the net force at each particle, *i.e.* the sum of equivariant pairwise forces and non-equivariant field effects. Hence, in this system, we say that *equivariance is entangled*.

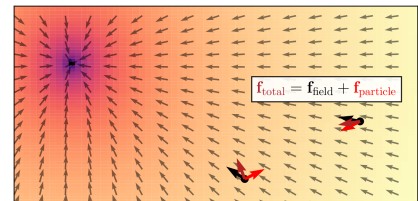

Figure 2: Two objects in a gravitational field. We only observe the total force exerted at each particle, *i.e.* the sum of equivariant pairwise particle forces and global field effects.

We now propose our architecture that disentangles local object interactions from global field effects. We model object interactions with local coordinate frame graph networks [24], and field effects with neural fields. During training, and given a multitude of input systems, neural fields will, in principle, be able to isolate

global from local effects, since only global effects are recurring phenomena. We hypothesize that field effects can be attributed to force fields, and therefore, our neural fields learn to discover *latent force fields*. The pipeline of our method is shown in Figure 3. Our inputs comprise augmented states $\{\mathbf{v}_i^t\}$ for trajectories of $N$ objects for $T$ timesteps. Since neural fields model global fields, they depend on absolute states. Thus, we feed the states of the trajectories $\mathbf{v}_i^t$ –or a subset of state variables– as input to a neural field that predicts latent forces $\mathbf{f}_i^t = \mathbf{f}(\mathbf{v}_i^t)$.

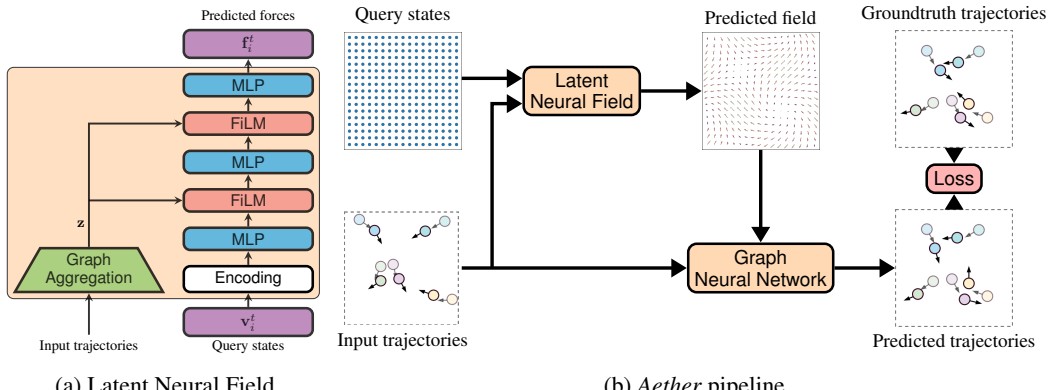

(a) Latent Neural Field        (b) *Aether* pipeline

Figure 3: The pipeline of our method, *Aether*. In the latent neural field (a), a graph aggregation module summarizes the input trajectories in a latent variable $\mathbf{z}$. Query states from input trajectories, alongside $\mathbf{z}$, are fed to a neural field that predicts a latent force field. In (b), a graph network integrates predicted forces with input trajectories to predict future trajectories. The graph aggregation module and the FiLM layers exist only in a dynamic field setting.

The predicted field forces can be now considered part of the node states, and further, they can be treated similarly to other state variables like velocities; as vectors, forces are unaffected by the action of translations, while they covariantly transform with rotations. Thus, moving onward, we can treat the problem setup as if we were once again back in the strict equivariance regime. We append the predicted forces $\mathbf{f}_i^t$ for each node-object $i$ and each timestep $t$ to the node states, and transform them to corresponding local coordinate frames, similarly to Equation (1). Namely, the force exerted on node $j$, expressed in the local coordinate frame of node $i$ is computed as: $\mathbf{f}_{j|i}^t = \mathbf{Q}^\top(\boldsymbol{\omega}_i^t)\mathbf{f}_j^t$. We feed the new local node states to a local coordinate frame graph network as follows:

$$\mathbf{h}_{j,i}^t = f_e\left(\left[\mathbf{v}_{j|i}^t, \boxed{\mathbf{f}_{j|i}^t}, \mathbf{v}_{i|i}^t, \boxed{\mathbf{f}_{i|i}^t}\right]\right) \tag{4}$$

$$\Delta\mathbf{x}_{i|i}^{t+1} = f_v\left(g_v\left(\left[\mathbf{v}_{i|i}^t, \boxed{\mathbf{f}_{i|i}^t}\right]\right) + C\sum_{j\in\mathcal{N}(i)}\mathbf{h}_{j,i}^t\right) \tag{5}$$

$$\mathbf{x}_i^{t+1} = \mathbf{x}_i^t + \mathbf{R}\left(\boldsymbol{\omega}_i^t\right)\cdot\Delta\mathbf{x}_{i|i}^{t+1}, \tag{6}$$

where $\mathbf{R}(\boldsymbol{\omega}_i^t) = \mathbf{Q}(\boldsymbol{\omega}_i^t) \oplus \mathbf{Q}(\boldsymbol{\omega}_i^t), C = 1/|\mathcal{N}(i)|$. The equations above are similar to Equations (2) and (3), with the addition of the highlighted parts that denote the predicted forces expressed at local coordinate frames. In practice, in most experiments, we closely follow [23, 14, 24] and formulate our model as a variational autoencoder [21, 37] with latent edge types. The exact details are presented in Appendix A.1.2.

## 3.2 Field discovery

Oftentimes, fields might not be directly observable for us to probe them at will and use them for supervision. For example, astronomical observations of solar systems and galaxies might not include black holes, yet we can observe their effects. Moreover, fields are often not even measurable or quantifiable, or they are defined implicitly. For instance, "social fields" that guide traffic, cannot be measured or defined explicitly, but we can safely assume they exist. Motivated by these observations, we design an architecture that performs *unsupervised field discovery*, while solving the surrogate supervised task of trajectory forecasting.

In this work, we aim to discover two different types of fields, which we term "static" and "dynamic" fields. Static fields refer to settings in which we have a single field shared throughout the whole dataset. On the other hand, dynamic fields refer to settings in which we have a different field for each input system, and consequently, fields also differ between train, validation, and test sets.

We now describe neural fields, used in this work, to model the underlying field effects. Neural fields depend on absolute states and predict latent force fields. When dealing with *static* fields, we use *unconditional* neural fields, *i.e.* neural fields that are functions only of the query states, as the field values are common across data samples. Note that unconditional neural fields are *not* functions of the input states; they will make the same predictions regardless of the inputs. In contrast, for *dynamic* fields, we use a *conditional* neural field, *i.e.* a neural field that also depends on a latent vector $\mathbf{z} \in \mathbb{R}^{D_z}$ that represents the underlying field. The latent $\mathbf{z}$ will be inferred from the input trajectories and can be thought of as representing unusual non-equivariant dynamics. We use $\mathbf{z}$ to explicitly condition the neural field, and thus, its general form is $\mathbf{f} : \mathbb{R}^d \times \mathrm{SO}(d) \times \mathbb{R}^d \times \mathbb{R}^{D_z} \to \mathbb{R}^d$, where $d \in \{2, 3\}$, depending on the setting.

During training, both for conditional and unconditional neural fields, we only sample the field at query states that coincide with the states of the input objects, since we only have supervision about their future trajectories there.

**Static fields**  We start with the description of unconditional neural fields used in static field settings, since conditional neural fields share the same backbone. First, we encode the query positions using Gaussian random Fourier features [46], as follows: $\gamma(\mathbf{p}) = [\cos(2\pi \mathbf{B}\mathbf{p}), \sin(2\pi \mathbf{B}\mathbf{p})]^\top$, where $\mathbf{p} \in \mathbb{R}^d$ are the query coordinates, and $\mathbf{B} \in \mathbb{R}^{\frac{D_c}{2} \times d}$ is a matrix with entries sampled from a Gaussian distribution, $\mathbf{B}_{kl} \sim \mathcal{N}(0, \sigma^2)$. The variance $\sigma^2$ can be chosen per task with a hyperparameter sweep.

We encode velocities using a simple linear layer $\zeta(\mathbf{u}) = \mathbf{W}_u \mathbf{u}$. For orientations, in $d = 3$ dimensions, we use a unit vector representation for each angle in $\boldsymbol{\omega} = (\theta, \phi, \psi)^\top$, $\hat{\boldsymbol{\omega}} = [\cos \boldsymbol{\omega}, \sin \boldsymbol{\omega}]^\top$. In $d = 2$ dimensions, we use the same encoding, except that we now have a single angle $\boldsymbol{\omega} = \theta$. Then, we use a linear layer to encode the orientation vectors, $\delta(\boldsymbol{\omega}) = \mathbf{W}_\omega \hat{\boldsymbol{\omega}}$. We finally concatenate the encoded positions, orientations, and velocities in a single vector that is being fed as input to the neural field. The neural field is a 3-layer MLP with SiLU [36] activations in-between, and outputs a latent force field, $\mathbf{f}(\mathbf{v}) = \mathrm{MLP}([\gamma(\mathbf{p}), \delta(\boldsymbol{\omega}), \zeta(\mathbf{u})])$.

**Dynamic fields**  The neural fields used to model the dynamic fields are conditioned on a latent vector representation $\mathbf{z} \in \mathbb{R}^{D_z}$ that describes prior knowledge about the underlying field, and are defined as $\mathbf{f}(\mathbf{v} \mid \mathbf{z})$. In our case, the latent representation should "summarize" the input graph such that it isolates only global effects from the field. To that end, we employ a simple global spatio-temporal attention mechanism, similar to Li et al. [25], that aggregates the input system in a latent vector representation. First, we define object embeddings $\mathbf{o}_i = \mathrm{GRU}(\mathbf{W}_g \mathbf{x}_i^{1:T})$, where $\mathbf{W}_g$ is a matrix used to linearly transform the inputs, and GRU is the Gated Recurrent Unit [7]. We also define temporal embeddings $\mathbf{t} = \mathrm{PE}(t)$, where PE are positional encodings [49]. Using these embeddings, we augment the input as $\mathbf{s}_i^t = [\mathbf{x}_i^t, \mathbf{o}_i] + \mathbf{t}$. The aggregation is then defined as follows:

$$\mathbf{z} = \sum_{i,t} \mathrm{softmax}(f_a(\mathbf{s}_i^t)) \cdot f_b(\mathbf{s}_i^t), \tag{7}$$

where $f_a : \mathbb{R}^{D_s} \to \mathbb{R}$, $f_b : \mathbb{R}^{D_s} \to \mathbb{R}^{D_z}$ are 2-layer MLPs with SiLU activations in-between.

After having obtained a latent vector representation $\mathbf{z}$ that summarizes the input system, we condition the neural field using FiLM [33]. We include FiLM layers after the first two linear layers of the neural field. The exact details are presented in Appendix A.1.1.

### 3.3 Approximate equivariance with global-local coordinate frames

Equivariant neural networks cannot capture non-local information, such as global field effects. In this work, we explicitly aim to discover these fields and disentangle them from local object interactions. An alternative, or rather complementary approach, would be to directly combine global and local information, following the recently proposed notion of *approximate equivariance* [51]. Starting from LoCS [24], we can integrate global information and still operate in local coordinate frames

by defining an auxiliary node-object corresponding to the global coordinate frame, *i.e.* an object positioned at the origin, and oriented to match the x-axis.

Similar to all objects in the system, the full state of the origin node $\mathcal{O}$ comprises the concatenation of its position and velocity, $\mathbf{x}_{\mathcal{O}} = [\mathbf{p}_{\mathcal{O}}, \mathbf{u}_{\mathcal{O}}]$. We use an "artificial" velocity that matches the $x$-axis in order to compute a non-degenerate frame. As such, we have $\mathbf{x}_{\mathcal{O}} = [\mathbf{0}, \hat{\mathbf{x}}]$. The origin state can be expressed in the local coordinate frame of the $i$-th object similarly to Equation (1), as follows:

$$\mathbf{v}_{\mathcal{O}|i}^{t} = \mathbf{R}_{i}^{t\top} \left[\mathbf{p}_{\mathcal{O}}^{t} - \mathbf{p}_{i}^{t}, \mathbf{u}_{\mathcal{O}}^{t}\right] = \mathbf{R}_{i}^{t\top} \left[-\mathbf{p}_{i}^{t}, \mathbf{u}_{\mathcal{O}}^{t}\right]. \tag{8}$$

Since graph networks are permutation equivariant, we need to explicitly distinguish between the origin node and other nodes. We circumvent that by augmenting each object's state with the origin node information expressed in local coordinate frames, extending Equations (2) and (3) to

$$\mathbf{h}_{j,i}^{t} = f_{e}\left(\left[\mathbf{v}_{j|i}^{t}, \mathbf{v}_{i|i}^{t}, \boxed{\mathbf{v}_{\mathcal{O}|i}^{t}}\right]\right), \tag{9}$$

$$\boldsymbol{\Delta}\mathbf{x}_{i|i}^{t+1} = f_{v}\left(g_{v}\left(\left[\mathbf{v}_{i|i}^{t}, \boxed{\mathbf{v}_{\mathcal{O}|i}^{t}}\right]\right) + \frac{1}{|\mathcal{N}(i)|}\sum_{j\in\mathcal{N}(i)}\mathbf{h}_{j,i}^{t}\right). \tag{10}$$

This approach pushes the information in the node states, and removes the need to add the origin node to the actual graph. We term this method *G-LoCS* (**G**lobal-**Lo**cal **C**oordinate Frame**S**). In practice, similar to Aether, we formulate G-LoCS as a variational autoencoder [21, 37] with latent edge types. The full details are presented in Appendix A.2. Finally, in practice, we integrate G-LoCS in Aether, since it can enhance the performance of our method.

## 4 Related work

**Equivariant graph networks**   The seminal works of [8, 9, 53] introduced equivariant convolutional neural networks and demonstrated effectiveness, robustness, and increased parameter efficiency. Recently, many works have proposed equivariant graph networks [43, 47, 12, 50, 41, 24, 5, 26, 18]. Walters et al. [50] propose rotationally equivariant continuous convolutions for trajectory prediction. Satorras et al. [41] propose a computationally efficient equivariant graph network that leverages invariant euclidean distances between node pairs. Kofinas et al. [24] introduce roto-translated local coordinate frames for all objects in an interacting system and propose equivariant local coordinate frame graph networks. Brandstetter et al. [5] generalize equivariant graph networks using steerable MLPs [47] and incorporate geometric and physical information in message passing. Equivariant graph networks differ from our work since they cannot capture non-local information, while our work disentangles equivariant local interactions from global effects and captures them both.

**Approximate equivariance**   Recently, a number of works has proposed to shift away from strict equivariance, in what Wang et al. [51] termed as *approximate equivariance*. Wang et al. [51] propose approximately equivariant networks for dynamical systems, by relaxing equivariance constraints in group convolutions and steerable convolutions. van der Ouderaa et al. [48] propose to relax strict equivariance by interpolating between equivariant and non-equivariant operations, using non-stationary kernels that also depend on the absolute input group element. Romero and Lohit [38] propose Partial G-CNNs that learn layer-wise partial equivariances from data. We note that even though approximately equivariant networks share similarities with our work, our notion of disentangled equivariance is conceptually different. That is because related work uses the term approximate equivariance to denote that equivariance is "broken" due to noise or imperfections, while our work disentangles the system dynamics that are actually equivariant, from the global field effects that are not, and in fact, might be unaffected by such transformations. Further, to the best of our knowledge, approximate equivariance has only been studied in the context of convolutional networks, not in the context of graph networks and interacting systems. Tangentially, Han et al. [15] propose subequivariant graph networks, and relax equivariance to subequivariance by considering external fields like gravity. However, they assume a priori known fields that do not require to be inferred by the model.

**Neural fields**   Neural fields have recently exploded in popularity in 3D computer vision, popularized by NeRF [29]. Since MLPs are universal function approximators [17], neural fields parameterized by MLPs can, in principle, encode continuous signals at arbitrary resolution. However, neural

networks can suffer from "spectral bias" [34, 2], *i.e.* they are biased to fit functions with low spatial frequency. To address this issue, a number of solutions have been proposed. Tancik et al. [46] leverage Neural Tangent Kernel (NTK) theory and propose Random Fourier Features (RFF), showing that they can overcome the spectral bias. They also show that RFF are a generalization of positional encodings, popularized in recent years in natural language processing by Transformers [49]. Concurrently, Sitzmann et al. [45] proposed SIREN, neural networks with sinusoidal activation functions. While neural fields have been used extensively in computer vision problems including 3D scene reconstruction [31, 28] and differentiable rendering [44, 29], they have not seen wide usage in dynamical systems. Notably, Raissi et al. [35] proposed Physics-Informed Neural Networks (PINNs), neural PDE solvers based on neural fields. Finally, Dupont et al. [11] and Zhuang et al. [56] propose generative models of neural fields.

## 5 Experiments

We evaluate our proposed method, *Aether*, on settings that include static as well as dynamic fields. First, we explore 2D charged particles that evolve under the effect of a static electrostatic field, as well as 3D particles that evolve under a Lorentz force field [10]. Then, we evaluate our method on a subset of inD [4] that contains a single location, and thus a static field as well. Finally, we explore 3D gravitational n-body problems [5] with dynamic fields. Our code, data, and models will be open-sourced online[1].

In most experiments, we compare our method against dNRI [14] and LoCS [24] , two state-of-the-art networks for sequence-to-sequence trajectory forecasting, as well as G-LoCS. DNRI [14] is a graph network operating in global coordinates, and is, in principle, able to uncover both the global and the local dynamics. It is formulated as a VAE [21, 37] with latent edge types and explicitly infers a latent graph structure. LoCS [24], on the other hand, operates in local coordinates, and is, thus, unable to uncover the global dynamics. Finally, G-LoCS is in principle able to model both local and global dynamics effectively. For all methods, we use their publicly available source code.

Our architecture and experimental setup closely follow Graber and Schwing [14], Kofinas et al. [24]. Unless specified differently, our neural field has a hidden size of 512. In charged particles and in n-body problems, we only use positions as input to the neural field, while in traffic scenes we also use orientations. The full implementation details are presented in Appendix A.1.2. In all settings, we report the mean squared error (MSE) of positions and velocities over time. Here we demonstrate indicative visualizations, and provide more extensive qualitative results in Appendix D.

For the Lorentz force field experiment, we use the official source code from ClofNet [10], and follow their exact setup. We compare our method against SE(3) Transformers [12], EGNN [41], and ClofNet [10]. We evaluate methods using the mean squared error between predicted and groundtruth positions. Since this setting is not a sequence-to-sequence task, we use a simplified network architecture *without* a VAE, and following baselines, we make sure that the number of parameters of our model is approximately equal to other methods. The full implementation details are presented in Appendix A.1.3.

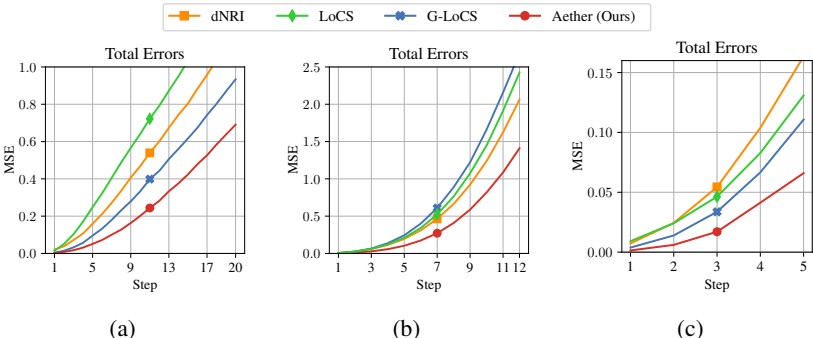

Figure 4: Results on (a) electrostatic field, (b) inD, and (c) gravity.

---

[1] https://github.com/mkofinas/aether

## 5.1 Electrostatic field

First, we study the effect of static fields, *i.e.* a single field across all train, validation, and test simulations. We extend the charged particles dataset from Kipf et al. [23] by adding a number of immovable sources. These sources act like regular particles, exerting forces on the observable particles, except we ignore any forces exerted to them, and keep their positions fixed. We use $M = 20$ "source" particles and $N = 5$ "observable" particles. We generate 50,000 simulations for training, 10,000 for validation and 10,000 for testing. Following Kipf et al. [23], each simulation lasts for 49 timesteps. During

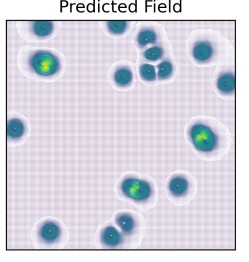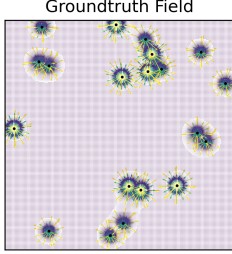

Predicted Field     Groundtruth Field

Figure 5: Learned Field (left) in electrostatic field setting compared to groundtruth (right).

inference, we use the first 29 steps as input and predict the remaining 20 steps. The full dataset details are presented in Appendix B.1.

We compare our method against dNRI, LoCS, and G-LoCS. We plot MSE in Figure 4a and $L_2$ errors in Figure 18, and visualize the learned field in Figure 5. We showcase predicted trajectories in Figure 9 in Appendix D.1. We observe that equivariant methods like LoCS perform poorly, while the approximately equivariant G-LoCS performs much better than equivariant and non-equivariant methods. Aether outperforms all other methods, demonstrating that it can *disentangle equivariance*. Furthermore, as shown in Figure 5, and Figure 10 in Appendix D.1.1, *Aether can effectively discover the underlying field*.

## 5.2 Lorentz force field

Du et al. [10] introduced a dataset of 3D charged particles evolving under the influence of a Lorentz force field. Each simulation contains 20 particles. We use the official source code and follow the exact experimental setup with Du et al. [10]. We show quantitative results in Table 1. Our method can clearly outperform all other methods by a large margin, reducing the error by $48.6\%$. We also note that our method has fewer parameters than ClofNet, and is thus more efficient.

Table 1: Position prediction MSE on Lorentz force field. Results marked with † were taken from ClofNet [10].

| Method | MSE ($\downarrow$) | No. parameters |
|---|---|---|
| GNN † | 0.0908 | 104,387 |
| SE(3) Transformer † [12] | 0.1438 | 1,763,134 |
| EGNN † [41] | 0.0368 | 134,020 |
| ClofNet † [10] | 0.0251 | 160,964 |
| Aether (ours) | **0.0129** | 132,822 |

## 5.3 Traffic scenes

Next, we study the effectiveness of static field discovery in traffic scenes. We use inD [4], a dataset with real-world traffic scenes that comprises trajectories of pedestrians, vehicles, and cyclists. We create a subset that contains scenes from a single location. The full dataset details are presented in Appendix B.2. We divide scenes into 18-step sequences; we use the first 6 time steps as input and predict the next 12 time steps. We plot MSE in Figure 4b and $L_2$ errors in Figure 19, and visualize the learned field in Figure 6, and in Figure 15 in Appendix D.2.1. Since the learned field is a function of positions and orientations, we only visualize it for 4 discrete orientations, namely the group $C_4 = \left\{0, \frac{\pi}{2}, \pi, \frac{3\pi}{2}\right\}$. We showcase predictions in Figure 11 in Appendix D.2. Again, Aether outperforms all other methods. The discovered field, while hard to interpret, shows high activations that coincide with road locations *and* directions, indicating that it can guide objects through the topology of the road network.

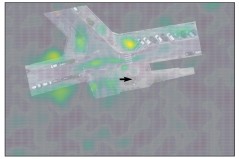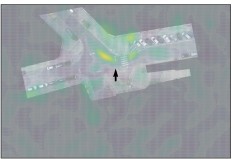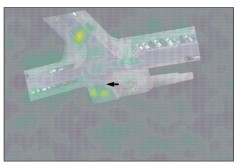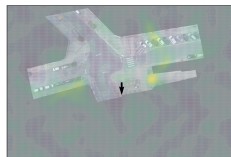

Figure 6: Discovered field on inD [4]. For clarity, we only visualize the field for discrete input orientations in $C_4 = \left\{0, \frac{\pi}{2}, \pi, \frac{3\pi}{2}\right\}$.

## 5.4  Gravitational field

We now study the task of dynamic field discovery, *i.e.* fields that are different across simulations. We extend the gravity dataset by Brandstetter et al. [5] by adding gravitational sources. We create a dataset of 50,000 simulations for training, 10,000 for validation and 10,000 for testing. We use $N = 5$ particles and $M = 1$ source. We set the masses of particles to $m_p = 1$, while the source's mass is $m_s = 10$. We generate trajectories of 49 timesteps. We use the first 44 steps as input and predict the remaining 5 steps. We plot the MSE in Figure 4c and $L_2$ errors in Figure 20. We observe that once again, Aether clearly outperforms other methods.

## 5.5  Ablation experiments

Table 2: (a) Ablation study on the importance of the learned field. (b) Ablation study on the importance of a sequential architecture. (c) Ablation study on the choice of equivariant GNN backbone.

| (a) Electrostatic field | | (b) Lorentz force field | | (c) Lorentz force field | |
|---|---|---|---|---|---|
| Method | MSE@10 ($\downarrow$) | Method | MSE ($\downarrow$) | Method | MSE ($\downarrow$) |
| Particle Oracle | 0.1847 | LoCS [24] | 0.0238 | EGNN [41] | 0.0368 |
| Force Oracle | 0.1883 | Aether | **0.0129** | EGNN+Aether | **0.0254** |
| Aether | 0.2015 | Parallel Aether | 0.0211 | | |

**Significance of discovered field**  In a simulated environment like the electrostatic field setting, we have access to the groundtruth fields and the sources that generate them. We leverage the simulator to study the significance of the discovered field in the task of trajectory forecasting, and establish an upper bound to our performance. To that end, we create two "oracle" models that have access to the groundtruth information, a *force oracle* and a *source oracle*. The force oracle is identical to Aether, but uses the groundtruth forces from the simulator instead of predicting them with a neural field. The source oracle assumes knowledge of the "field sources". Thus, there is no longer need for disentanglement, and the problem is strictly equivariant again. We include the sources as virtual nodes in the graph, add include edges from the sources to the particles. We describe this oracle in detail in Appendix A.3. We show the MSE in Table 2a and plot the MSE and $L_2$ errors in Figure 22 in Appendix E.5. We observe that Aether closely follows the two oracle models, and is on par, for roughly 10 steps. This demonstrates that the discovered field is almost as helpful as the groundtruth.

**Learning the global field separately**  Our architecture connects the neural field with the graph network sequentially, *i.e.* the output of neural field is given as input to the graph network. We believe that this is integral for effective field discovery, as well as overall modelling, since it enables the graph network to learn to isolate local interactions from the observed net dynamics. We test this hypothesis with an ablation study, in which we connect the neural field and the graph network in parallel. The two networks are now working independently, and we only add their predictions at the output. We term this model *Parallel Aether*. We provide implementation details in Appendix A.4. We perform the experiment on the Lorentz field setting, and show results in Table 2b. We also compare both methods against LoCS, as it is a common backbone in both methods, to demonstrate the performance gain due to the discovered field. We can see that even though the parallel architecture boosts performance, it is clearly not as effective as the sequential approach, which verifies our hypothesis.

Table 3: Ablation study on using conditional neural fields for static fields. Experiment on Lorentz force field setting.

| Method | MSE ($\downarrow$) | No. parameters | Inference Time |
|---|---|---|---|
| LoCS [24] | 0.0238 | 130,307 | 0.0033 |
| Aether | 0.0129 | 132,822 | 0.0037 |
| Conditional Aether | 0.0131 | 142,807 | 0.0047 |

**Choice of equivariant network**    Our method is agnostic to the choice of equivariant graph network; we expect that it would be beneficial for a number of strictly equivariant networks. To test this hypothesis, we combine our method with EGNN [41]. We start from the velocity formulation of EGNN and modify the message and velocity equations to incorporate the predicted forces for each node. We describe the model in detail in Appendix A.5. We train and evaluate this method on the Lorentz force field setting, and report the results in Table 2c. EGNN combined with Aether reduces the error by 30.9%, compared to a vanilla EGNN, which enhances our hypothesis. We further include comparisons with more equivariant and non-equivariant graph networks in Appendix C.

**Conditional neural fields for static settings**    Conditional neural fields generalize unconditional neural fields, and could, in principle, be used to learn static fields. In that case, the neural field should learn to ignore the latent vector, since the generated field should be identical regardless of the input system; we expect its performance to match the unconditional field. This, however, can come at the cost of increased training and inference time, as well as redundant computational resources and model parameters. We verify this hypothesis with an ablation study on the Lorentz force field, where we train and evaluate our method using a conditional field. In Table 3, we report the MSE, as well as the training time per minibatch, the inference time, and the number of parameters for each model. While the conditional model performs almost on par with the original unconditional model, this comes at the cost of 27% higher inference time and 9,985 more parameters. We conclude that the unconditional neural field is the preferred choice when there is expert knowledge that the field at hand is a static field. In the absence of such knowledge, *e.g.* on an exploratory analysis for underlying fields, then the conditional neural field would be preferable.

## 6   Conclusion

In this work, we introduced *Aether*, a method that discovers global fields in interacting systems. We propose neural fields to discover latent force fields, and infer them from the dynamics alone. Furthermore, we disentangle global fields from local object interactions, and combine neural fields with equivariant graph networks to learn the systems. We show that our method can accurately discover the underlying fields in a range of settings with static and dynamic fields, and effectively use them to forecast future trajectories. To the best of our knowledge, *Aether* is the first work that discovers fields in interacting systems, and the first that is able to model systems with equivariant interactions and global fields. We hope that this work will inspire the community and bootstrap a line of works that explores field discovery, *since fields are omnipresent in all scientific tasks*.

**Limitations**    In this work, we have only considered fields that do not react to the observable environment. While this setting is often true, in other scenarios, active fields might be crucial for effective modelling of the system dynamics. Furthermore, in the dynamic field setting, we assume that the input trajectories are descriptive enough to summarize the field we are trying to discover. While this hypothesis often holds, it might not always be true. Future work can explore these very interesting research directions.

## Acknowledgments

The project is funded by the NWO LIFT grant 'FLORA'.

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

# A  Implementation details

## A.1  Aether

Here we present the full Aether architecture. We first describe the details of the neural field used for field discovery, and then we describe our graph network formulated as a variational autoencoder [21, 37].

### A.1.1  Neural field

In its general form, the neural field takes as inputs query states $\mathbf{v}$ that comprise positions $\mathbf{p} \in \mathbb{R}^d$, orientations $\boldsymbol{\omega} \in \mathrm{SO}(d)$, and velocities $\mathbf{u} \in \mathbb{R}^d$, as well as a latent code $\mathbf{z} \in \mathbb{R}^{D_z}$ used to condition the field, and predicts latent forces at the query states. Thus, it is defined as $\mathbf{f} : \mathbb{R}^d \times \mathrm{SO}(d) \times \mathbb{R}^d \times \mathbb{R}^{D_z} \to \mathbb{R}^d$, where $d \in \{2, 3\}$. Depending on the task at hand, we can omit the latent code $\mathbf{z}$, *e.g.* if we are modelling a static field, or the orientations $\boldsymbol{\omega}$, if we have prior knowledge that the field is independent to them.

**Encoding positions**  We encode the query positions using Gaussian random Fourier features [46], $\gamma(\mathbf{p}) = [\cos(2\pi\mathbf{B}\mathbf{p}), \sin(2\pi\mathbf{B}\mathbf{p})]^\top$, where $\mathbf{B} \in \mathbb{R}^{\frac{D_c}{2} \times d}$ is a matrix with entries sampled from a Gaussian distribution, $\mathbf{B}_{kl} \sim \mathcal{N}(0, \sigma^2)$. Throughout the experiments, and unless otherwise specified, we use a unit variance $\sigma^2 = 1$, and $\frac{D_c}{2} = 256$. Thus, the encoded positions have a dimension of 512.

**Encoding orientations**  In 3 dimensions, for the orientations $\boldsymbol{\omega} = (\theta, \phi, \psi)^\top$, we follow [24], and use the angles of the velocity vectors as a proxy. We represent each angle in $\boldsymbol{\omega}$ as unit vector, $\hat{\boldsymbol{\omega}} = [\cos\boldsymbol{\omega}, \sin\boldsymbol{\omega}]^\top$, and encode them with a linear layer $\delta(\hat{\boldsymbol{\omega}}) = \mathbf{W}_\omega \hat{\boldsymbol{\omega}}$, where $\mathbf{W}_\omega \in \mathbb{R}^{D_c \times |\hat{\boldsymbol{\omega}}|}$. In 2 dimensions, we use the same encoding, except that we now have a single angle $\boldsymbol{\omega} = \theta$.

**Encoding velocities**  For velocities, we simply encode them using a linear layer $\zeta(\mathbf{u}) = \mathbf{W}_u \mathbf{u}$, where $\mathbf{W}_u \in \mathbb{R}^{D_c \times d}$. Finally, we concatenate the encoded positions, orientations, and velocities in a single vector before we feed them as input to the neural field.

**Latent code**  The latent code $\mathbf{z}$ "summarizes" the input graph such that it isolates global field effects. We employ a simple global spatio-temporal attention mechanism, similar to Li et al. [25], that aggregates the input system in a latent vector representation. First, we define object embeddings $\mathbf{o}_i = \mathrm{GRU}\big(\mathbf{W}_g \mathbf{x}_i^{1:T}\big)$, where $\mathbf{W}_g \in \mathbb{R}^{D_o \times d}$ is a matrix used to linearly transform the inputs, and GRU is the Gated Recurrent Unit [7]. We also define temporal embeddings $\mathbf{t} = \mathrm{PE}(t)$, where PE are positional encodings [49], defined as:

$$\mathrm{PE}(t)_{2i} = \sin\Big(t/10000^{2i/D_s}\Big), \tag{11}$$

$$\mathrm{PE}(t)_{2i+1} = \cos\Big(t/10000^{2i/D_s}\Big), \tag{12}$$

where $i$ is the $i$-th dimension.

The aggregation is then defined as follows:

$$\mathbf{z} = \sum_{i,t} \mathrm{softmax}\big(f_a\big(\mathbf{s}_i^t\big)\big) \cdot f_b\big(\mathbf{s}_i^t\big), \quad \text{with} \quad \mathbf{s}_i^t = \big[\mathbf{x}_i^t, \mathbf{o}_i\big] + \mathbf{t}, \tag{13}$$

where $f_a : \mathbb{R}^{D_s} \to \mathbb{R}, f_b : \mathbb{R}^{D_s} \to \mathbb{R}^{D_z}$ are 2-layer MLPs with SiLU activations [36] in-between. They can be summarized as:

$$f_a := \{\mathrm{Linear}(D_s, D_z) \to \mathrm{SiLU} \to \mathrm{Linear}(D_z, 1)\}, \tag{14}$$
$$f_b := \{\mathrm{Linear}(D_s, D_z) \to \mathrm{SiLU} \to \mathrm{Linear}(D_z, D_z)\}. \tag{15}$$

In all experiments, we use $D_o = 512$, $D_z = 512$, and $D_s = D_o + 2d = 516$ in $d = 2$ dimensions, or $D_s = 518$ in $d = 3$ dimensions.

**Neural field conditioning** We condition the neural field using FiLM [33]. Following the implementation details of FiLM, in practice, we use the following equation for a FiLM layer:

$$\mathbf{h}' := \text{FiLM}(\mathbf{h}, \mathbf{z}) = (1 + \alpha(\mathbf{z})) \odot \mathbf{h} + \beta(\mathbf{z}), \tag{16}$$

where $\mathbf{h}$ is the encoded input in the first FiLM layer, or the conditioned input in subsequent FiLM layers, and $\alpha : \mathbb{R}^{D_z} \to \mathbb{R}^{D_h}, \beta : \mathbb{R}^{D_z} \to \mathbb{R}^{D_h}$ are MLPs. This equation deviates slightly from Section 2, since it predicts the residual of a multiplicative modulation. This approach can be beneficial during the early stages of training, since it initially defaults to an identity transformation for zero-initialized weights, while the alternative can "zero out" the network outputs. For both $\alpha$ and $\beta$ we use 2-layer MLPs with SiLU activations in-between.

$$\alpha := \{\text{Linear}(D_z, D_h) \to \text{SiLU} \to \text{Linear}(D_h, D_h)\} \tag{17}$$
$$\beta := \{\text{Linear}(D_z, D_h) \to \text{SiLU} \to \text{Linear}(D_h, D_h)\}. \tag{18}$$

Unless specified otherwise, in all experiments, we use $D_h = 512$.

**Full neural field** The full neural field is a 3-layer MLP with SiLU [36] activations in-between, and FiLM layers after the first two linear layers, and outputs a latent force field. The neural field can be summarized as:

$$\mathbf{f}(\mathbf{v} \mid \mathbf{z}) = \{\text{Linear} \to \text{FiLM} \to \text{SiLU} \to \text{Linear} \to \text{FiLM} \to \text{SiLU} \to \text{Linear}\}. \tag{19}$$

### A.1.2 Aether as a variational autoencoder

Here we present our graph network architecture that closely follows Graber and Schwing [14], Kofinas et al. [24]. The model is formulated as a variational autoencoder [21, 37] with latent edge types that infers a latent graph structure. The encoder is tasked with predicting interactions between object pairs, while the decoder uses the sampled graph structure to make predictions. As mentioned in Section 3.3, our full architecture also integrates G-LoCS, *i.e.* the augmented node states include the predicted forces exerted at the target node, as well as the state of the auxiliary origin-node, expressed in the local frame of the target node.

**Encoder** Equations (20) to (22) describe the message passing steps of our graph network. In these equations, we process each timestep independently. Then, in Equations (23) and (24) we compute the evolution of edge embeddings over time with LSTMs [16], and in Equations (25) and (26) we estimate the posterior and the learned prior over our edges.

$$\mathbf{h}_{j,i}^{(1),t} = f_e^{(1)}\left(\left[\mathbf{v}_{j|i}^t, \mathbf{f}_{j|i}^t, \mathbf{v}_{i|i}^t, \mathbf{f}_{i|i}^t, \mathbf{v}_{\mathcal{O}|i}^t\right]\right) \tag{20}$$

$$\mathbf{h}_i^{(1),t} = f_v^{(1)}\left(g_v^{(1)}\left(\left[\mathbf{v}_{i|i}^t, \mathbf{f}_{i|i}^t, \mathbf{v}_{\mathcal{O}|i}^t\right]\right) + \frac{1}{|\mathcal{N}(i)|}\sum_{j \in \mathcal{N}(i)} \mathbf{h}_{j,i}^{(1),t}\right) \tag{21}$$

$$\mathbf{h}_{j,i}^{(2),t} = f_e^{(2)}\left(\left[\mathbf{h}_i^{(1),t}, \mathbf{h}_{j,i}^{(1),t}, \mathbf{h}_j^{(1),t}\right]\right) \tag{22}$$

$$\mathbf{h}_{(j,i),\text{prior}}^t = \text{LSTM}_{\text{prior}}\left(\mathbf{h}_{j,i}^{(2),t}, \mathbf{h}_{(j,i),\text{prior}}^{t-1}\right) \tag{23}$$

$$\mathbf{h}_{(j,i),\text{enc}}^t = \text{LSTM}_{\text{enc}}\left(\mathbf{h}_{j,i}^{(2),t}, \mathbf{h}_{(j,i),\text{enc}}^{t+1}\right) \tag{24}$$

$$p_\phi\left(\mathbf{z}^t|\mathbf{x}^{1:t}, \mathbf{z}^{1:t-1}\right) = \text{softmax}\left(f_{\text{prior}}\left(\mathbf{h}_{(j,i),\text{prior}}^t\right)\right) \tag{25}$$

$$q_\phi\left(\mathbf{z}_{j,i}^t|\mathbf{x}\right) = \text{softmax}\left(f_{\text{enc}}\left(\left[\mathbf{h}_{(j,i),\text{prior}}^t, \mathbf{h}_{(j,i),\text{enc}}^t\right]\right)\right) \tag{26}$$

The functions $f_e^{(1)}, f_v^{(1)}, f_e^{(2)}, g_v^{(1)}, f_{\text{prior}}, f_{\text{enc}}$ denote MLPs.

**Decoder** The decoder samples $\mathbf{z}_{(j,i)}^t$ using Gumbel-Softmax [27, 19]. The following equations formalize a message passing scheme performed for the current timestep, and another one performed for the hidden node states. Both are used to update the hidden node states, and to make predictions for the next timestep. As mentioned in Section 2, we make predictions in the local coordinate frame

of each node. Thus, we perform an inverse transformation for each node to transform the predictions back to the global coordinate frame.

$$\mathbf{m}_{j,i}^t = \sum_k z_{(j,i),k}^t f^k \left( \left[ \mathbf{v}_{j|i}^t, \mathbf{f}_{j|i}^t, \mathbf{v}_{i|i}^t, \mathbf{f}_{i|i}^t, \mathbf{v}_{\mathcal{O}|i}^t \right] \right) \tag{27}$$

$$\mathbf{m}_i^t = f_v^{(3)} \left( g_v^{(3)} \left( \left[ \mathbf{v}_{i|i}^t, \mathbf{f}_{i|i}^t, \mathbf{v}_{\mathcal{O}|i}^t \right] \right) + \frac{1}{|\mathcal{N}(i)|} \sum_{j \in \mathcal{N}(i)} \mathbf{m}_{j,i}^t \right) \tag{28}$$

$$\mathbf{h}_{j,i}^t = \sum_k z_{(j,i),k}^t g^k \left( \left[ \mathbf{h}_j^t, \mathbf{h}_i^t \right] \right) \tag{29}$$

$$\mathbf{n}_i^t = \frac{1}{|\mathcal{N}(i)|} \sum_{j \in \mathcal{N}(i)} \mathbf{h}_{(j,i)}^t \tag{30}$$

$$\mathbf{h}_i^{t+1} = \text{GRU} \left( \left[ \mathbf{n}_i^t, \mathbf{m_i}^t \right], \mathbf{h}_i^t \right) \tag{31}$$

$$\boldsymbol{\mu}_i^{t+1} = \mathbf{x}_i^t + \mathbf{R}_i^t \cdot f_v^{(4)} \left( \mathbf{h}_i^{t+1} \right) \tag{32}$$

$$p(\mathbf{x}_i^{t+1} | \mathbf{x}^{1:t}, \boldsymbol{z}^{1:t}) = \mathcal{N} \left( \boldsymbol{\mu}_i^{t+1}, \sigma^2 \mathbf{I} \right) \tag{33}$$

Our GRU [1] is identical to the one used in [23].

**Training**  Our full VAE model is trained by minimizing the negative Evidence Lower Bound (ELBO), which comprises the reconstruction loss of the predicted trajectories (positions and velocities) and the KL divergence.

$$\mathcal{L}(\phi, \theta) = \mathbb{E}_{q_\phi(\boldsymbol{z}|\mathbf{x})}[\log p_\theta(\mathbf{x}|\boldsymbol{z})] - \text{KL}[q_\phi(\boldsymbol{z}|\mathbf{x})||p_\phi(\boldsymbol{z}|\mathbf{x})] \tag{34}$$

Following Graber and Schwing [14], the reconstruction loss and the KL divergence take the following form:

$$\mathbb{E}_{q_\phi(\boldsymbol{z}|\mathbf{x})}[\log p_\theta(\mathbf{x}|\boldsymbol{z})] = -\sum_i \sum_t \frac{||\mathbf{x}_i^t - \boldsymbol{\mu}_i^t||}{2\sigma^2} + \frac{1}{2} \log \left( 2\pi\sigma^2 \right), \tag{35}$$

$$\text{KL}[q_\phi(\boldsymbol{z}|\mathbf{x})||p_\phi(\boldsymbol{z}|\mathbf{x})] = \sum_{t=1}^{T} \left( \mathbb{H}(q_\phi(\boldsymbol{z}_{ji}^t|\mathbf{x})) - \sum_{\boldsymbol{z}_{ji}^t} q_\phi(\boldsymbol{z}_{ji}^t|\mathbf{x}) \log p_\phi(\boldsymbol{z}_{ji}^t|\mathbf{x}^{1:t}, \boldsymbol{z}^{1:t-1}) \right), \tag{36}$$

where $\mathbb{H}$ denotes the entropy operator. In all experiments, we set the variance $\sigma^2 = 10^{-5}$.

We train Aether using Adam [21]. Unless stated otherwise, in all experiments, we use a learning rate of $5e-4$.

### A.1.3  Aether architecture in Lorentz force field setting

The Lorentz force field setting, proposed by [10] uses only a single timestep as input and the task is to predict the positions for a single timestep in the future. Thus, we have to modify our architecture for this setting. This way, we also ensure a fairer comparison with other methods. We do not use an NRI [22] or dNRI [14] backbone in this setting.

$$\mathbf{h}_{j,i}^{(1)} = f_e^{(1)} \left( \left[ \mathbf{v}_{j|i}, \mathbf{f}_{j|i}, \mathbf{v}_{i|i}, \mathbf{f}_{i|i}, q_i q_j, \|\mathbf{r}_{j,i}\|_2 \right] \right) \tag{37}$$

$$\mathbf{h}_i^{(1)} = f_v^{(1)} \left( g_v \left( \left[ \mathbf{v}_{i|i}, \mathbf{f}_{i|i} \right] \right) + \frac{1}{|\mathcal{N}(i)|} \sum_{j \in \mathcal{N}(i)} \mathbf{h}_{j,i}^{(1)} \right) \tag{38}$$

$$\mathbf{h}_{j,i}^{(l)} = f_e^{(l)} \left( \left[ \mathbf{h}_i^{(l-1)}, \mathbf{h}_{j,i}^{(l-1)}, \mathbf{h}_j^{(l-1)} \right] \right) \tag{39}$$

$$\mathbf{h}_i^{(l)} = f_v^{(l)} \left( \mathbf{h}_i^{(l-1)} + \frac{1}{|\mathcal{N}(i)|} \sum_{j \in \mathcal{N}(i)} \mathbf{h}_{j,i}^{(l)} \right) \tag{40}$$

$$\hat{\mathbf{p}}_i = \mathbf{p}_i + \mathbf{R}_i \cdot f_o \left( \mathbf{h}_i^L \right) \tag{41}$$

with $l \in \{2, \ldots, L\}$. We use 4 layers, *i.e.* $L = 4$. The functions $f_e^{(l)}$ denote 2-layer MLPs with SiLU [36] activations after each layer. The functions $f_v^{(l)}$ denote 2-layer MLPs with SiLU activations in-between, and doubling the dimensionality in-between. Finally, $g_v$ denotes a linear layer, while $f_o$ denotes a 3-layer MLP with SiLU activations in-between. Following [10], we use a hidden dimension of 64. For this experiment, we use a learning rate of $1e - 3$.

$$f_v^{(l)} = \{\text{Linear} \to \text{SiLU} \to \text{Linear}\} \tag{42}$$

$$f_e^{(l)} = \{\text{Linear} \to \text{SiLU} \to \text{Linear} \to \text{SiLU}\} \tag{43}$$

$$f_o = \{\text{Linear} \to \text{SiLU} \to \text{Linear} \to \text{SiLU} \to \text{Linear}\} \tag{44}$$

For the neural field, we use the input positions and velocities as input, as well as the particle charges, *i.e.* $\mathbf{f} = f(\mathbf{p}, \mathbf{u}, q)$. We do not use any input encoding for positions or velocities in this setting, but we use an embedding for the charges that maps them to 16 dimensions. We concatenate the charge embeddings with positions and velocities and feed them as input to the neural field. The neural field is a 3-layer MLP with SiLU activations in-between. We use a hidden dimension of 32 in the neural field.

$$f = \{\text{Linear} \to \text{SiLU} \to \text{Linear} \to \text{SiLU} \to \text{Linear}\} \tag{45}$$

## A.2 G-LoCS

**Encoder**

$$\mathbf{h}_{j,i}^{(1),t} = f_e^{(1)}\left(\left[\mathbf{v}_{j|i}^t, \mathbf{v}_{i|i}^t, \mathbf{v}_{\mathcal{O}|i}^t\right]\right) \tag{46}$$

$$\mathbf{h}_i^{(1),t} = f_v^{(1)}\left(g_v^{(1)}\left(\left[\mathbf{v}_{i|i}^t, \mathbf{v}_{\mathcal{O}|i}^t\right]\right) + \frac{1}{|\mathcal{N}(i)|}\sum_{j \in \mathcal{N}(i)} \mathbf{h}_{j,i}^{(1),t}\right) \tag{47}$$

$$\mathbf{h}_{j,i}^{(2)} = f_e^{(2)}\left(\left[\mathbf{h}_i^{(1),t}, \mathbf{h}_{j,i}^{(1),t}, \mathbf{h}_j^{(1),t}\right]\right) \tag{48}$$

$$\mathbf{h}_{(j,i),\text{prior}}^t = \text{LSTM}_{\text{prior}}\left(\mathbf{h}_{j,i}^{(2),t}, \mathbf{h}_{(j,i),\text{prior}}^{t-1}\right) \tag{49}$$

$$\mathbf{h}_{(j,i),\text{enc}}^t = \text{LSTM}_{\text{enc}}\left(\mathbf{h}_{j,i}^{(2),t}, \mathbf{h}_{(j,i),\text{enc}}^{t+1}\right) \tag{50}$$

$$p_\phi\left(\mathbf{z}^t | \mathbf{x}^{1:t}, \mathbf{z}^{1:t-1}\right) = \text{softmax}\left(f_{\text{prior}}\left(\mathbf{h}_{(j,i),\text{prior}}^t\right)\right) \tag{51}$$

$$q_\phi\left(\mathbf{z}_{j,i}^t | \mathbf{x}\right) = \text{softmax}\left(f_{\text{enc}}\left(\left[\mathbf{h}_{(j,i),\text{prior}}^t, \mathbf{h}_{(j,i),\text{enc}}^t\right]\right)\right) \tag{52}$$

**Decoder**

$$\mathbf{m}_{j,i}^t = \sum_k z_{(j,i),k}^t f^k\left(\left[\mathbf{v}_{j|i}^t, \mathbf{v}_{i|i}^t, \mathbf{v}_{\mathcal{O}|i}^t\right]\right) \tag{53}$$

$$\mathbf{m}_i^t = f_v^{(3)}\left(g_v^{(3)}\left(\left[\mathbf{v}_{i|i}^t, \mathbf{v}_{\mathcal{O}|i}^t\right]\right) + \frac{1}{|\mathcal{N}(i)|}\sum_{j \in \mathcal{N}(i)} \mathbf{m}_{j,i}^t\right) \tag{54}$$

$$\mathbf{h}_{j,i}^t = \sum_k z_{(j,i),k}^t g^k\left(\left[\mathbf{h}_j^t, \mathbf{h}_i^t\right]\right) \tag{55}$$

$$\mathbf{n}_i^t = \frac{1}{|\mathcal{N}(i)|}\sum_{j \in \mathcal{N}(i)} \mathbf{h}_{(j,i)}^t \tag{56}$$

$$\mathbf{h}_i^{t+1} = \text{GRU}\left(\left[\mathbf{n}_i^t, \mathbf{m_i}^t\right], \mathbf{h}_i^t\right) \tag{57}$$

$$\boldsymbol{\mu}_i^{t+1} = \mathbf{x}_i^t + \mathbf{R}_i^t \cdot f_v^{(4)}\left(\mathbf{h}_i^{t+1}\right) \tag{58}$$

$$p(\mathbf{x}_i^{t+1} | \mathbf{x}^{1:t}, \mathbf{z}^{1:t}) = \mathcal{N}\left(\boldsymbol{\mu}_i^{t+1}, \sigma^2 \mathbf{I}\right) \tag{59}$$

## A.3 Source oracle

The *source oracle* modifies LoCS [24] to use virtual nodes. Since graph networks are permutation invariant, we cannot just include the sources as nodes of the graph and perform message passing,

as the network would not be able to distinguish particles from sources. Thus, we treat the sources separately in the message passing so that the network can identify them. More specifically, we introduce a new message function that computes field source $\rightarrow$ particle messages. Furthermore, we introduce a separate aggregation function in the update step that only aggregates the messages from field sources. We denote the set of field sources as $\mathcal{S}$. The state of a field source $s \in \mathcal{S}$ is denoted as $\mathbf{v}_s$, while the same state expressed in the local coordinate frame of node $i$ is denoted as $\mathbf{v}_{s|i}$. The source oracle graph network is defined as follows:

$$\mathbf{h}_{j,i}^t = f_e\left(\left[\mathbf{v}_{j|i}^t, \mathbf{v}_{i|i}^t\right]\right), \tag{60}$$

$$\mathbf{h}_{s,i}^t = f_s\left(\left[\mathbf{v}_{s|i}^t, \mathbf{v}_{i|i}^t\right]\right), \tag{61}$$

$$\boldsymbol{\Delta}\mathbf{x}_{i|i}^{t+1} = f_v\left(g_v\left(\mathbf{v}_{i|i}^t\right) + \frac{1}{|\mathcal{N}(i)|}\sum_{j\in\mathcal{N}(i)}\mathbf{h}_{j,i}^t + \frac{1}{|\mathcal{S}|}\sum_{j\in\mathcal{S}}\mathbf{h}_{s,i}^t\right), \tag{62}$$

where $f_s$ is an MLP. We predict future states for all the "observable" particles, but not for the "source" particles.

### A.4 Parallel aether architecture

$$\mathbf{h}_{j,i}^{(1)} = f_e^{(1)}\left(\left[\mathbf{v}_{j|i}, \mathbf{v}_{i|i}, q_i q_j, \|\mathbf{r}_{j,i}\|_2\right]\right) \tag{63}$$

$$\mathbf{h}_i^{(1)} = f_v^{(1)}\left(g_v\left(\mathbf{v}_{i|i}\right) + \frac{1}{|\mathcal{N}(i)|}\sum_{j\in\mathcal{N}(i)}\mathbf{h}_{j,i}^{(1)}\right) \tag{64}$$

$$\mathbf{h}_{j,i}^{(l)} = f_e^{(l)}\left(\left[\mathbf{h}_i^{(l-1)}, \mathbf{h}_{j,i}^{(l-1)}, \mathbf{h}_j^{(l-1)}\right]\right) \tag{65}$$

$$\mathbf{h}_i^{(l)} = f_v^{(l)}\left(\mathbf{h}_i^{(l-1)} + \frac{1}{|\mathcal{N}(i)|}\sum_{j\in\mathcal{N}(i)}\mathbf{h}_{j,i}^{(l)}\right) \tag{66}$$

$$\hat{\mathbf{p}}_i = \mathbf{p}_i + \mathbf{R}_i \cdot f_o\left(\mathbf{h}_i^L\right) + \boxed{\mathbf{f}_i} \tag{67}$$

### A.5 Aether with EGNN backbone

Our method is agnostic to the choice of equivariant graph network; we expect that it would be beneficial for a number of strictly equivariant networks. To test this hypothesis, we combine EGNN [41] with our method; starting from the velocity formulation of EGNN, we modify the message and velocity equations to incorporate the predicted forces for each node, as follows:

$$\mathbf{m}_{j,i} = \phi_e\left(\mathbf{h}_i^l, \mathbf{h}_j^l, \left\|\mathbf{p}_j^l - \mathbf{p}_i^l\right\|_2^2, a_{j,i}, \boxed{\mathbf{f}_i}, \boxed{\mathbf{f}_j}\right), \tag{68}$$

$$\mathbf{u}_i^{l+1} = \phi_v\left(\mathbf{h}_i^l, \boxed{\mathbf{f}_i}\right)\mathbf{u}_i^l + C\sum_{j\neq i}\left(\mathbf{p}_j^l - \mathbf{p}_i^l\right)\cdot\phi_x(\mathbf{m}_{j,i}), \tag{69}$$

$$\mathbf{p}_i^{l+1} = \mathbf{p}_i^l + \mathbf{u}_i^{l+1}, \tag{70}$$

$$\mathbf{m}_i = \sum_{j\in\mathcal{N}(i)}\mathbf{m}_{ji}, \tag{71}$$

$$\mathbf{h}_i^{l+1} = \phi_h\left(\mathbf{h}_i^l, \mathbf{m}_i\right). \tag{72}$$

The remaining EGNN components remain unaltered.

### A.6 Computing resources

All experiments were performed on single GPUs. We used 2 different GPU models, namely the Nvidia RTX 2080 Ti, and Nvidia GTX 1080 Ti. Our source code was written in PyTorch [32], version 1.4.0, and CUDA 10.0.

# B  Dataset details

## B.1  Electrostatic field

Kipf et al. [23] introduced a dataset of interacting charged particles. Charged particles interact via electrostatic Coulomb forces. We assume a set of $N$ particles, and each particle has a position $\mathbf{p}_i^t \in \mathbb{R}^D$ and a charge $q_i \in \mathbb{R}$. The force $\mathbf{f}_{j,i}^t$ exerted from particle $j$ to particle $i$ is computed as follows:

$$\mathbf{p}_{j,i}^t = \mathbf{p}_j^t - \mathbf{p}_i^t, \tag{73}$$

$$\hat{\mathbf{p}}_{j,i}^t = \frac{\mathbf{p}_{j,i}^t}{\left\| \mathbf{p}_{j,i}^t \right\|}, \tag{74}$$

$$\mathbf{f}_{j,i}^t = C \cdot q_i q_j \frac{\hat{\mathbf{p}}_{j,i}^t}{\left\| \mathbf{p}_{j,i}^t \right\|^2}. \tag{75}$$

Since forces only depend on positions at the current timestep, in the following equations, we omit the time indices to reduce clutter. The total force exerted at particle $i$ is:

$$\mathbf{f}_i = \sum_{j=1, j \neq i}^{N} \mathbf{f}_{j,i} = C \cdot q_i \sum_{j=1, j \neq i}^{N} q_j \frac{\hat{\mathbf{p}}_{j,i}}{\left\| \mathbf{p}_{j,i} \right\|^2}. \tag{76}$$

The electric field is a vector field, whose value at the test position $\mathbf{p}_i$ assumes a positive test charge $q_i = 1$, and is defined as:

$$\mathbf{E}_{j,i} = \frac{\mathbf{f}_{j,i}}{q_i} = C \cdot q_j \frac{\hat{\mathbf{p}}_{j,i}}{\left\| \mathbf{p}_{j,i} \right\|^2}, \tag{77}$$

$$\mathbf{E}_i = \sum_{j=1, j \neq i}^{N} \mathbf{E}_{j,i} = C \cdot \sum_{j=1, j \neq i}^{N} q_j \frac{\hat{\mathbf{p}}_{j,i}}{\left\| \mathbf{p}_{j,i} \right\|^2}. \tag{78}$$

Our first experiment aims to study the effect of static fields, *i.e.* a single field across all train, validation, and test simulations. We extend the charged particles dataset by adding a number of immovable sources. Overall, these sources act like regular particles, exerting forces on the observable particles, except we ignore any forces exerted to them, and fix their positions and velocities to zero. We use $N = 5$ "observable" particles and $M = 20$ "source" particles. In all experiments, we assume unit charges, $q_i = \pm 1$, and $C = 1$. The probabilities of positive or negative charges are equal. Then, the forces and the electric field can be simplified as:

$$\mathbf{f}_{j,i} = \text{sign}(q_i q_j) \frac{\hat{\mathbf{p}}_{j,i}}{\left\| \mathbf{p}_{j,i} \right\|^2}, \tag{79}$$

$$\mathbf{E}_i = \sum_{j=1, j \neq i}^{N} \text{sign}(q_j) \frac{\hat{\mathbf{p}}_{j,i}}{\left\| \mathbf{p}_{j,i} \right\|^2}. \tag{80}$$

The net force exerted at a particle $i \in \{1, \dots, N\}$ is computed as:

$$\mathbf{f}_i = \sum_{j=1, j \neq i}^{N+M} \mathbf{f}_{j,i} = \underbrace{\sum_{j=1, j \neq i}^{N} \mathbf{f}_{j,i}}_{\text{particles}} + \underbrace{\sum_{j=N+1}^{N+M} \mathbf{f}_{j,i}}_{\text{field}} \tag{81}$$

Following Satorras et al. [41], Fuchs et al. [12], Kofinas et al. [24], we remove virtual borders that cause elastic collisions. We generate a dataset of 50,000 simulations for training, 10,000 for validation and 10,000 for testing. The datasets contains *only* the positions and velocities for the "observable" particles, while the field sources are only used for visualization. Following Kipf et al. [23], each simulation lasts for 49 timesteps. During inference, we use the first 29 steps as input and predict the remaining 20 steps.

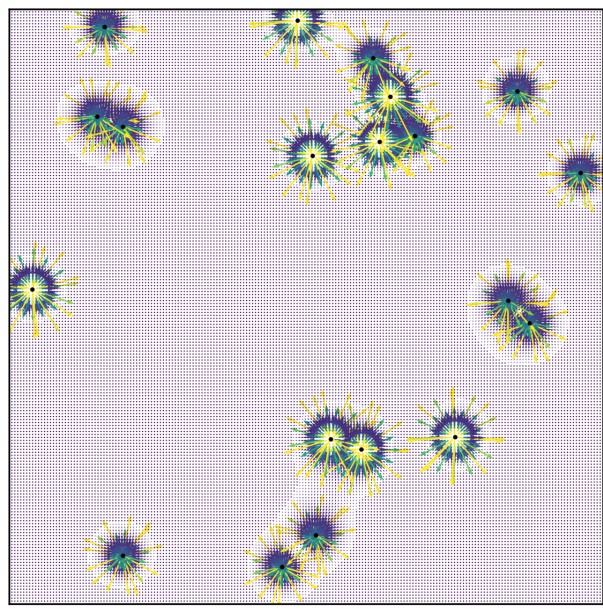

Figure 7: Visualization of the static field in the electrostatic field setting

## B.2 Traffic scenes - inD

InD [4] is a real-world traffic scenes dataset that comprises trajectories of pedestrians, vehicles, and cyclists. It contains 33 recordings, recorded at 4 different locations in Aachen, Germany. We hypothesize that discovering a latent traffic force field will be beneficial for trajectory forecasting in traffic scenes. For simplicity, we focus on static field discovery in traffic scenes. We create a subset that contains scenes from a single location. Namely, we choose "Frankenburg, Aachen", since it is the location with most interactions in the dataset. The subset corresponds to 12 recordings; we use 8 for training, 2 for validation, and 2 for testing. We follow a similar experimental setting with Graber and Schwing [14], Kofinas et al. [24]. We divide each scene into 18-step sequences. We use the first 6 time steps as input and predict the next 12 time steps.

## B.3 Gravitational n-body dataset

In this experiment, we study the influence of dynamic fields, *i.e.* fields that are different across simulations. Similar to the electrostatic field setting, we extend the gravitational n-body dataset by Brandstetter et al. [5] by adding gravitational sources. The equation that describes the forces is similar to Equation (75). Namely, we have

$$\mathbf{f}_{j,i}^t = C \cdot m_i m_j \frac{\hat{\mathbf{p}}_{j,i}^t}{\left\|\mathbf{p}_{j,i}^t\right\|^2},$$
(82)

where $m_i, m_j$ are the particle masses. We create a dataset of 50,000 simulations for training, 10,000 for validation and 10,000 for testing. We use $N = 5$ particles and $M = 1$ source. We set the masses of particles to $m_p = 1$, while the source has a mass of $m_s = 10$. Similarly to the electrostatic field experiment, the datasets contains *only* the positions and velocities for the "observable" particles, while the field source is only used for visualization. We generate trajectories of 49 timesteps. We use the first 44 timesteps as input and predict the remaining 5 steps. All other dataset details are identical to Brandstetter et al. [5].

### B.3.1 2D gravitational n-body dataset

We also experiment with a smaller variant of the dynamic gravitational fields, using a 2D setting. We create a dataset of 5,000 simulations for training, 1,000 for validation and 1,000 for testing. All other

dataset details are the same with the full 3D dataset. We report results in Figure 21. We showcase predicted trajectories in Appendix D.3, and the learned fields in Appendix D.3.1.

## C  Extra experiments

### C.1  Alternative equivariant network backbones

To further test the applicability of our method, we combine it with different equivariant graph network backbones. First, we combine our method with GMN [18]. Similar to Equations (4) and (5), we concatenate the predicted forces for each node with the message $\mathbf{Z}_{ji}$ in $[\mathbf{Z}_{ji}, \mathbf{f}_i, \mathbf{f}_j]$. The formulation above is further motivated by SGNN [15], which extends GMN by including gravity as an external force term, as well as object-aware information (see appendix A.3 in [15] for a comparison). In our case, we replace the gravity term with the predicted forces per node. Thus, instead of $[\mathbf{Z}_{ji}, \mathbf{g}]$, we have $[\mathbf{Z}_{ji}, \mathbf{f}_i, \mathbf{f}_j]$.

We train and evaluate GMN on the Lorentz force field setting. We then add the force terms using the formulation above. We show the results in the table below. Indeed, using Aether greatly enhances the performance of GMN, which further enhances our hypothesis.

Table 4: Ablation study on the choice of equivariant GNN backbone. Position prediction MSE on Lorentz force field.

| Method | MSE ($\downarrow$) |
| --- | --- |
| GMN [18] | 0.0365 |
| GMN+Aether (ours) | **0.0261** |

Next, we integrate our method in EqMotion [55], a recent equivariant method with state-of-the-art performance on trajectory forecasting. We incorporate Aether in EqMotion by treating the predicted forces as geometric features similar to velocities. Namely, after computing the forces for each object at each timestep, we compute the magnitudes of the force vectors and the force angle sequence, *i.e.* the angles between forces in consecutive timesteps. We concatenate these quantities to the existing features for the feature initialization step.

We train and evaluate EqMotion and Aether with EqMotion on inD [4] following our experimental setup. We report the results in Figure 8. We see that Aether is beneficial even for a state-of-the-art trajectory forecasting method, which further strengthens our claims.

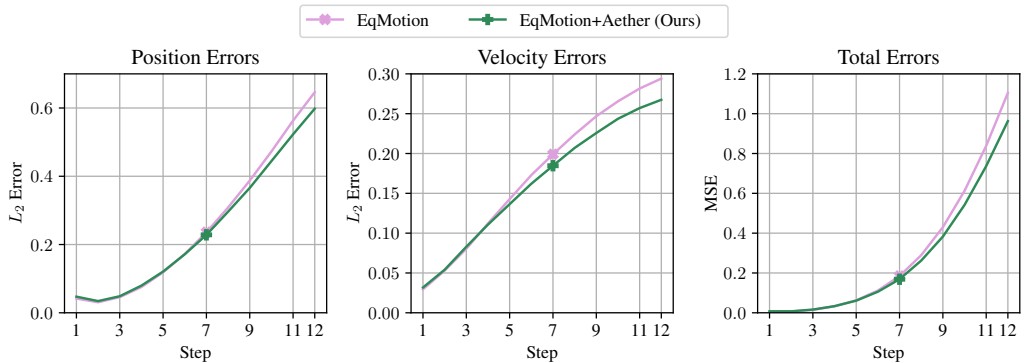

Figure 8: Ablation study on the choice of equivariant GNN backbone. Results on inD.

### C.2  Non-equivariant network with neural field

Our ability to capture global components with a neural field stems from our overall architecture, which promotes disentanglement. Using a graph network that respects the underlying symmetries and has an inductive bias towards using local interactions, *i.e.* any equivariant graph network, allows

the neural field to "solve for" the global components, by "subtracting" the local interactions from the observable net effects.

The choice of an equivariant network is crucial here; a non-equivariant graph network like NRI [23] or dNRI [14] would merely gather "redundant" information from the neural field. We demonstrate this mathematically in the following equations for the static field, in which we first compute the force $\mathbf{f}$ at a target position $\mathbf{p}$, and then compute the node embedding using equations from NRI/dNRI, including the forces.

$$\mathbf{f} = f(\mathbf{p}) = \text{MLP}_1(\mathbf{p}) \tag{83}$$
$$\mathbf{h} = g(\mathbf{p}, \mathbf{u}, \mathbf{f}) = \text{MLP}_2([\mathbf{p}, \mathbf{u}, \mathbf{f}]) \tag{84}$$

We can see that the node embeddings depend on positions twice, one explicit and one through another MLP, in an architecture similar to a concatenated residual connection. In this case, we do not expect the neural field to isolate global forces, or to be helpful for future forecasting. We test this hypothesis with an ablation experiment on the electrostatic field setting, by combing our neural field with dNRI, instead of an equivariant network. In Table 5, we report the MSE at the final prediction timestep, i.e. MSE@20. We can see that adding a neural field to a non-equivariant network does not enhance performance, and in fact, it results in performance degradation, which enhances our hypothesis.

Table 5: Ablation study on the suitability of non-equivariant networks with Aether. Combining dNRI –a non-equivariant graph network– with a neural field does not enhance performance. Results on the electrostatic field setting.

| Method | MSE@20 ($\downarrow$) |
|---|---|
| dNRI [14] | 1.20 |
| dNRI+Aether | 1.37 |
| Aether | **0.69** |

### C.3 Choice of conditioning mechanism

FiLM [33] is used in the dynamic field setting to condition neural fields. To examine its influence on performance, we perform an ablation study on the 3D gravitational setting, where we replace FiLM layers with conditioning by concatenation, a very simple and successful conditioning mechanism. We term this model *Concat Aether*, and report the results in Table 6. We see that conditioning by concatenation underperforms, scoring almost on par with LoCS. This is perhaps expected, since concatenation is a rather weak form of conditioning, and our task is very challenging. On the other hand, FiLM is a powerful mechanism and is able to condition effectively.

Table 6: Ablation study on the choice of conditioning mechanism. Results on the 3D gravitational field setting.

| Method | MSE@5 ($\downarrow$) |
|---|---|
| LoCS [24] | 0.1308 |
| Aether | **0.0660** |
| Concat Aether | 0.1474 |

# D Qualitative results

## D.1 Electrostatic field

Figure 9 shows qualitative results on the electrostatic field setting.

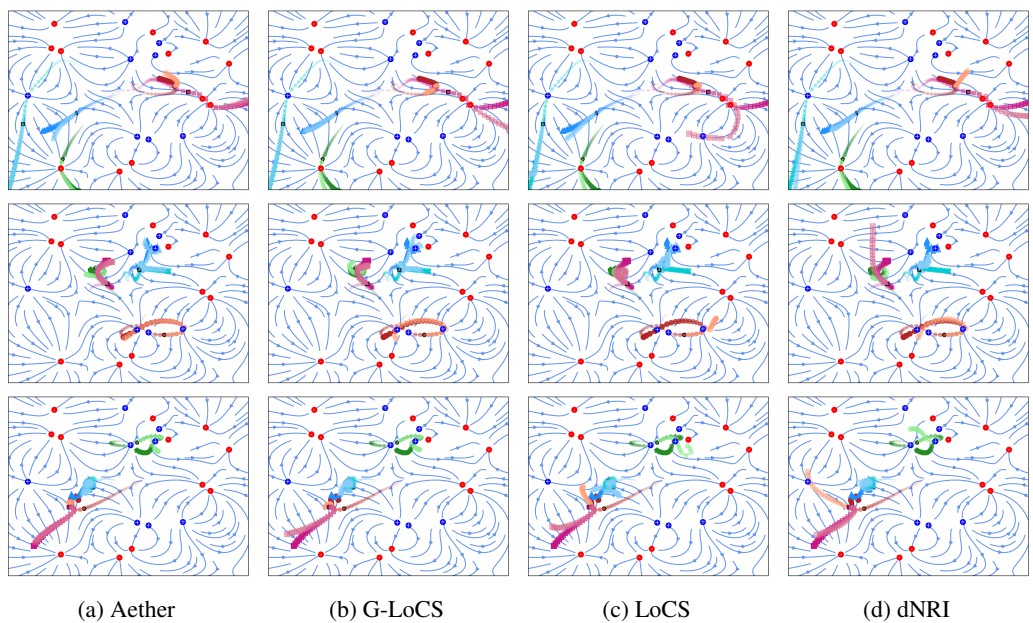

|  |  |  |  |
|---|---|---|---|
| (a) Aether | (b) G-LoCS | (c) LoCS | (d) dNRI |

Figure 9: Predictions on the electrostatic field setting. Lighter colors indicate predictions, and darker colors indicate the groundtruth. Predictions start where markers have black edges. Markers get bigger and more opaque as trajectories evolve in time. The background streamplots indicate the groundtruth field, and are not given as input to the networks. Similarly, the blue ⊕ markers and the red ⊖ markers, are merely shown for illustrative purposes, indicating the charges of the field sources, and are not given as input to the networks. *Best viewed in color.*

### D.1.1 Discovered electrostatic field

In Figure 10 we visualize the discovered electrostatic field compared to the groundtruth one.

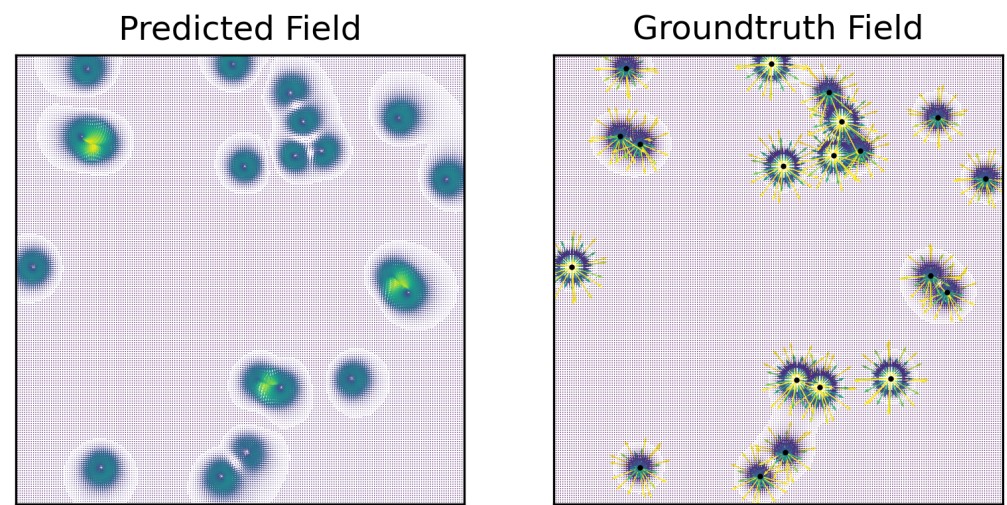

Figure 10: Learned Field (left) in electrostatic field setting compared to groundtruth (right).

## D.2  InD

Figure 11 shows qualitative results of our method on inD [4]. Figures 12, 13 and 14 show qualitative results for G-LoCS, LoCS, and dNRI, respectively.

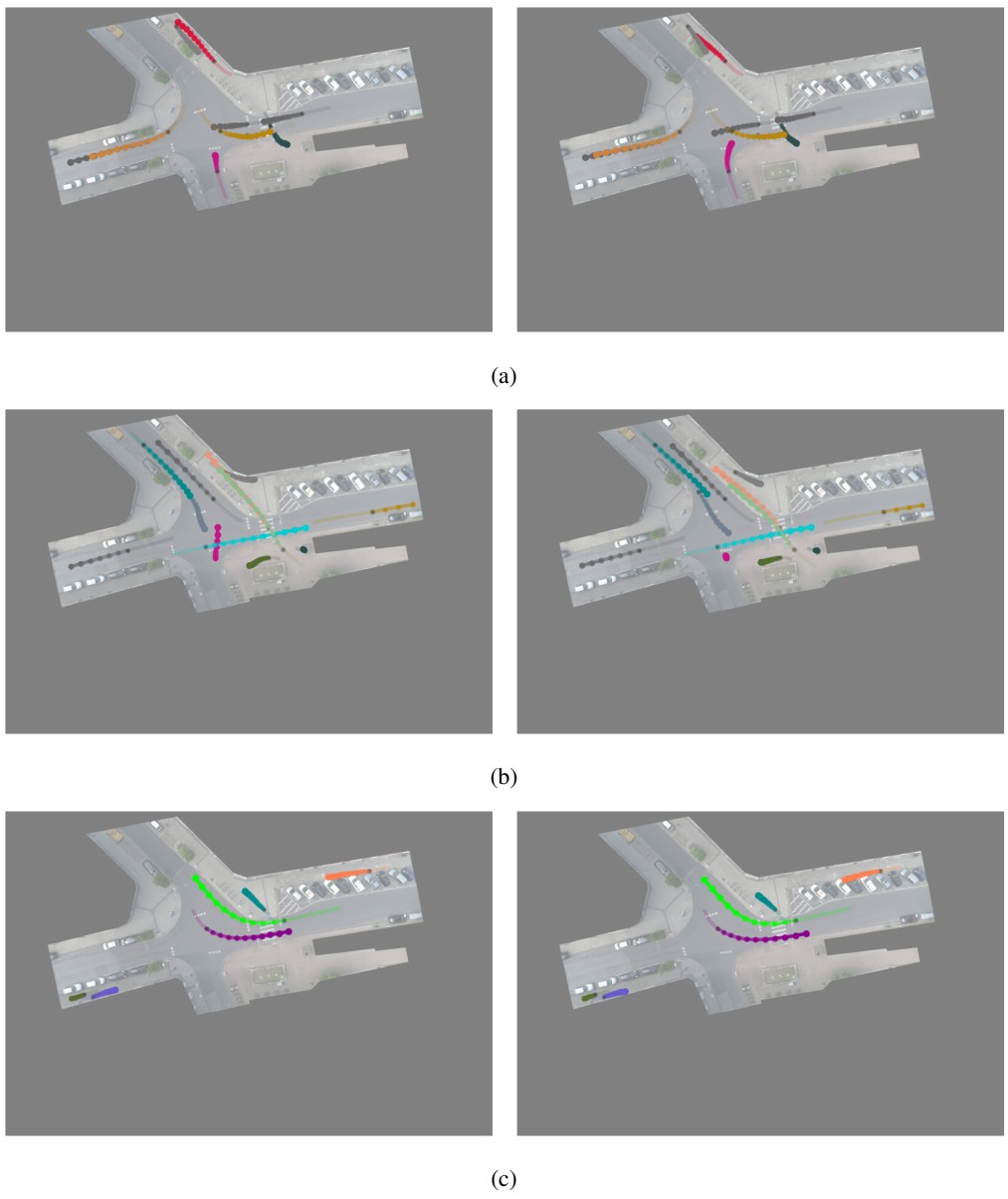

(a)

(b)

(c)

Figure 11: Aether predictions (right) on inD, compared to groundtruth (left). Predictions start where markers are colored black. *Best viewed in color.*

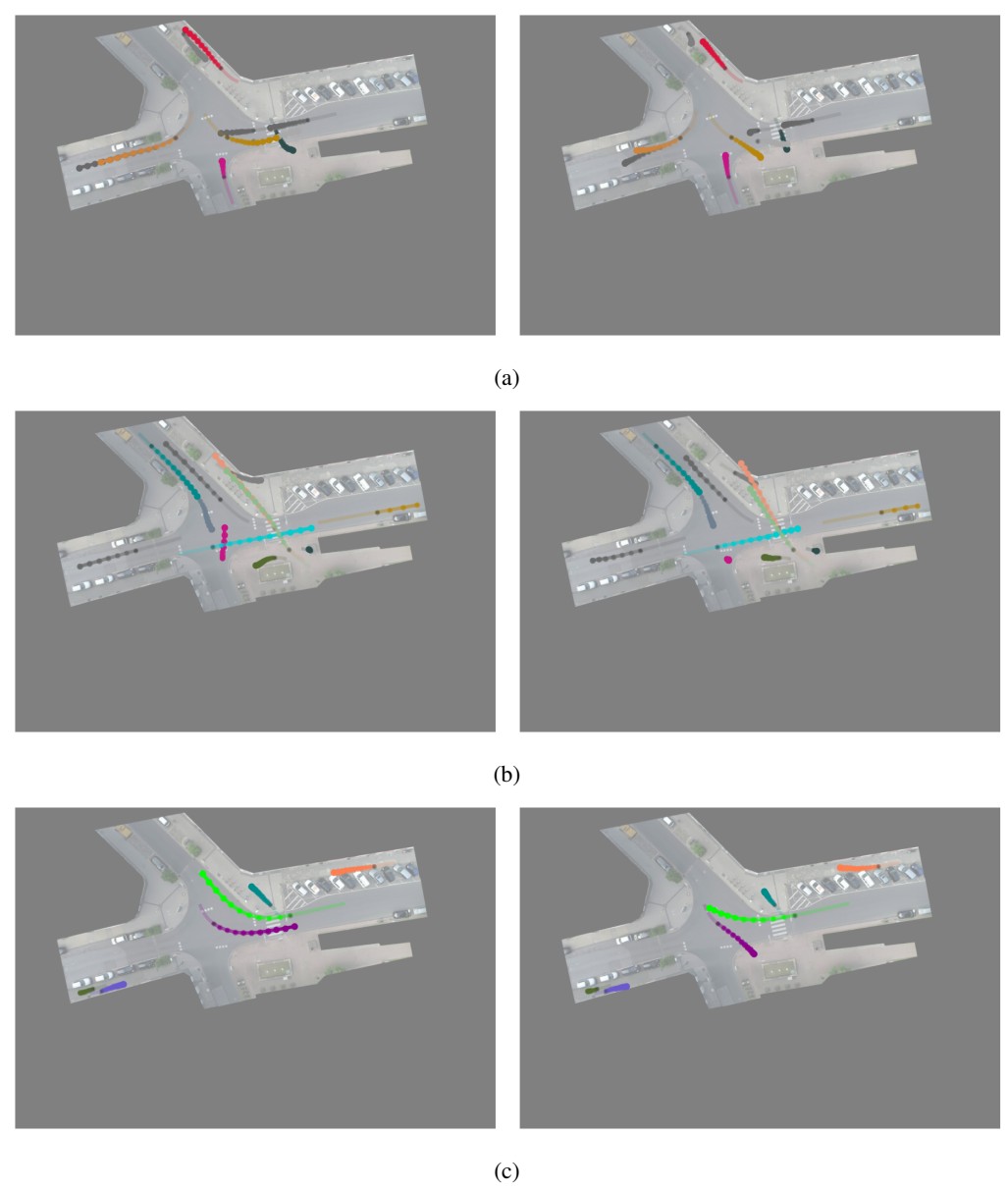

(a)

(b)

(c)

Figure 12: G-LoCS predictions (right) on inD, compared to groundtruth (left). Predictions start where markers are colored black. *Best viewed in color.*

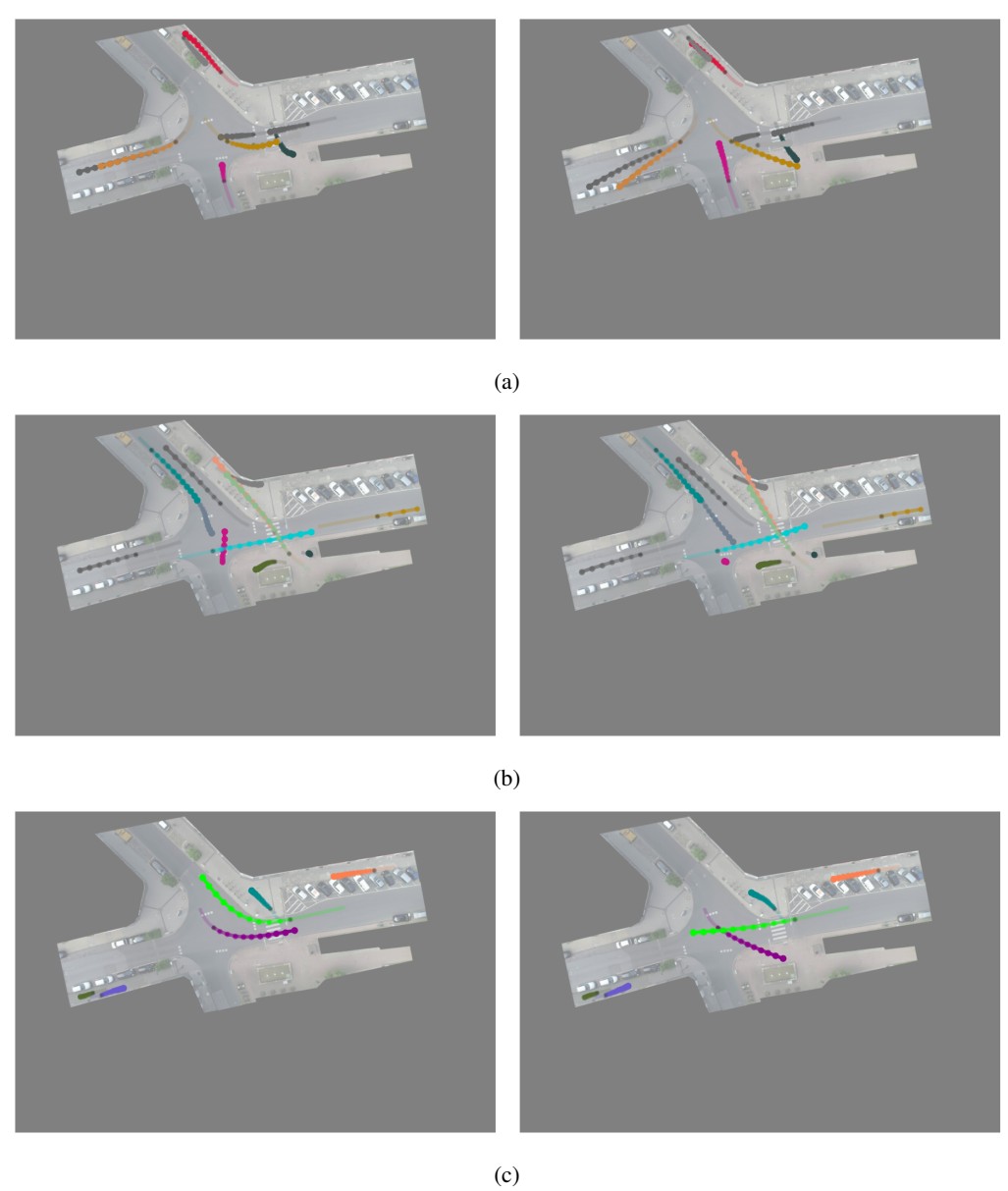

(a)

(b)

(c)

Figure 13: LoCS predictions (right) on inD, compared to groundtruth (left). Predictions start where markers are colored black. *Best viewed in color.*

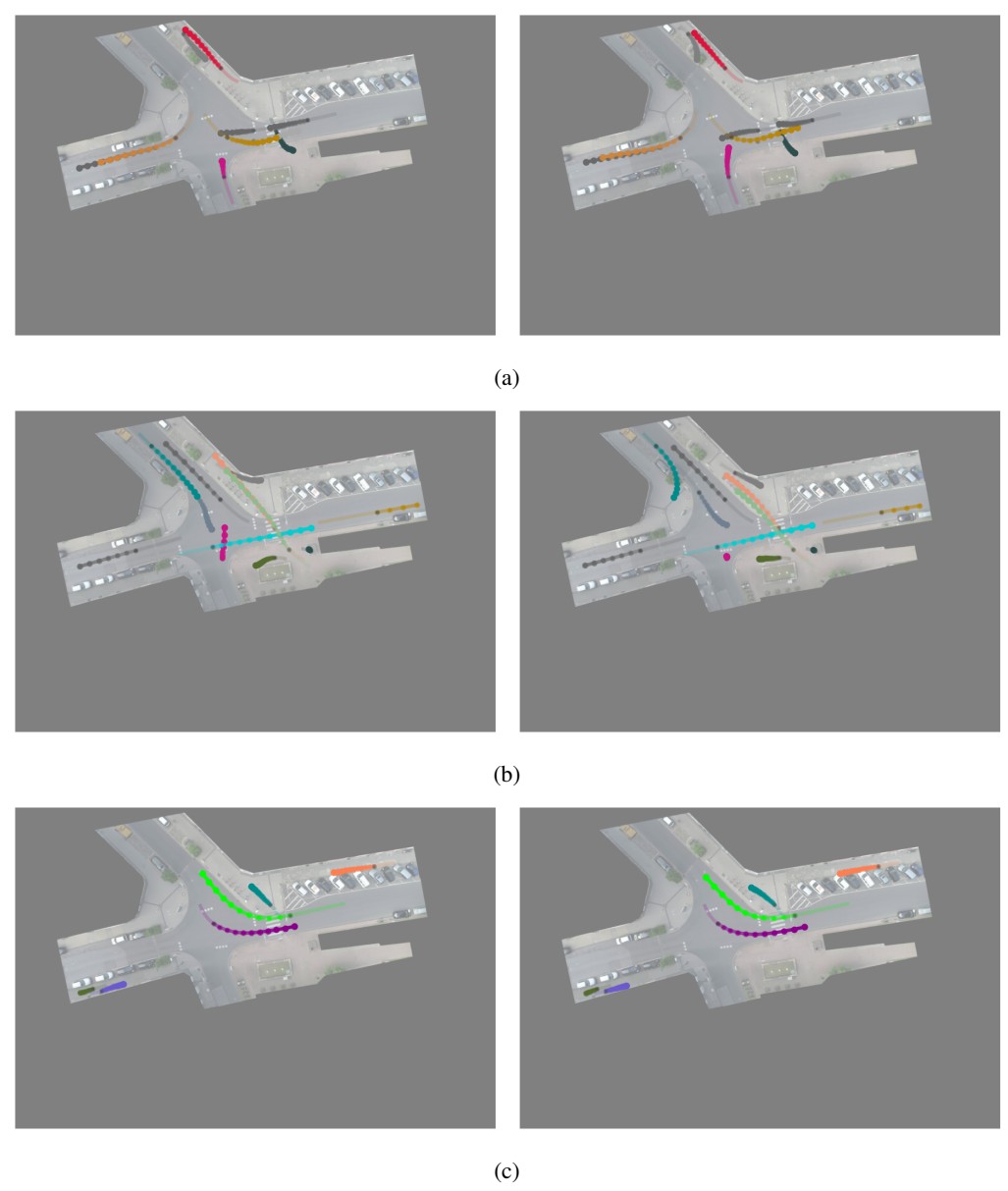

(a)

(b)

(c)

Figure 14: dNRI predictions (right) on inD, compared to groundtruth (left). Predictions start where markers are colored black. *Best viewed in color.*

### D.2.1   Discovered traffic force field

In Figure 15 we visualize the discovered traffic force field on inD. In the supplementary material, we provide video visualizations of the learned field, with input orientations evolving over time.

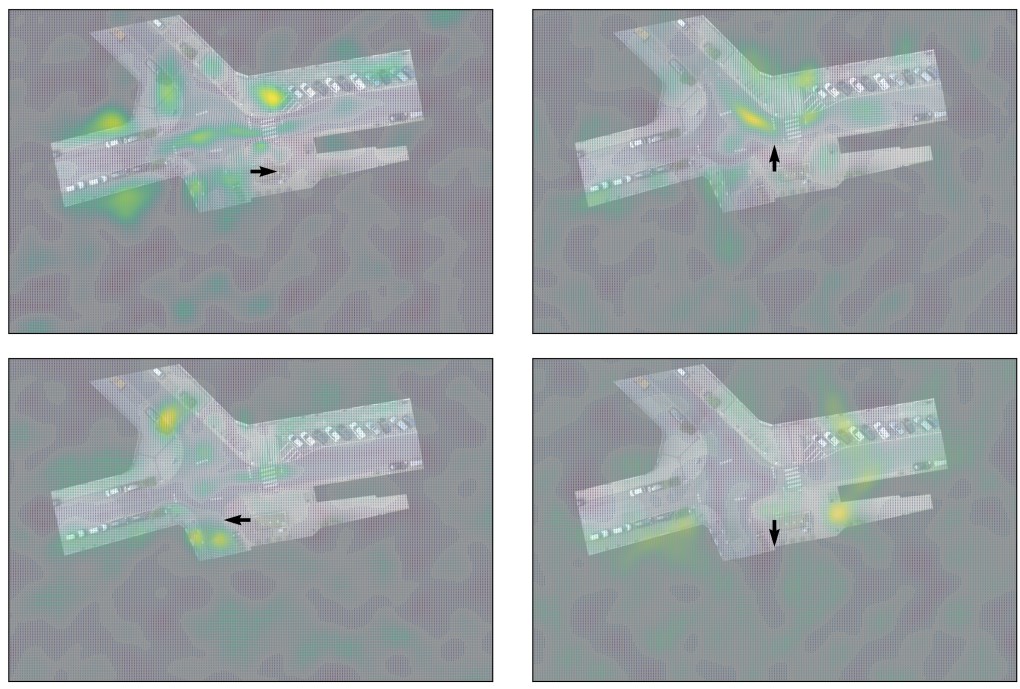

Figure 15: Discovered field on inD [4]. For simplicity, we only visualize the field for discrete input orientations in $C_4 = \left\{0, \frac{\pi}{2}, \pi, \frac{3\pi}{2}\right\}$. *Best viewed in color.*

### D.3  2D gravity

Figure 16 shows qualitative results on the 2D gravitational n-body problem.

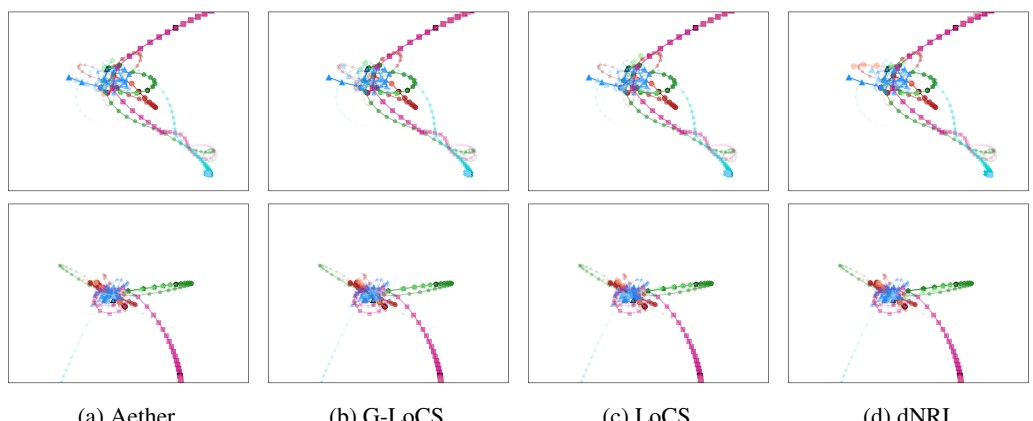

(a) Aether       (b) G-LoCS       (c) LoCS       (d) dNRI

Figure 16: Predictions on gravity. Lighter colors indicate predictions, and darker colors indicate the groundtruth. Predictions start where markers have black edges. Markers get bigger as trajectories evolve. *Best viewed in color.*

#### D.3.1  Discovered 2D gravitational fields

Figure 17 shows examples of discovered fields compared to the groundtruth ones.

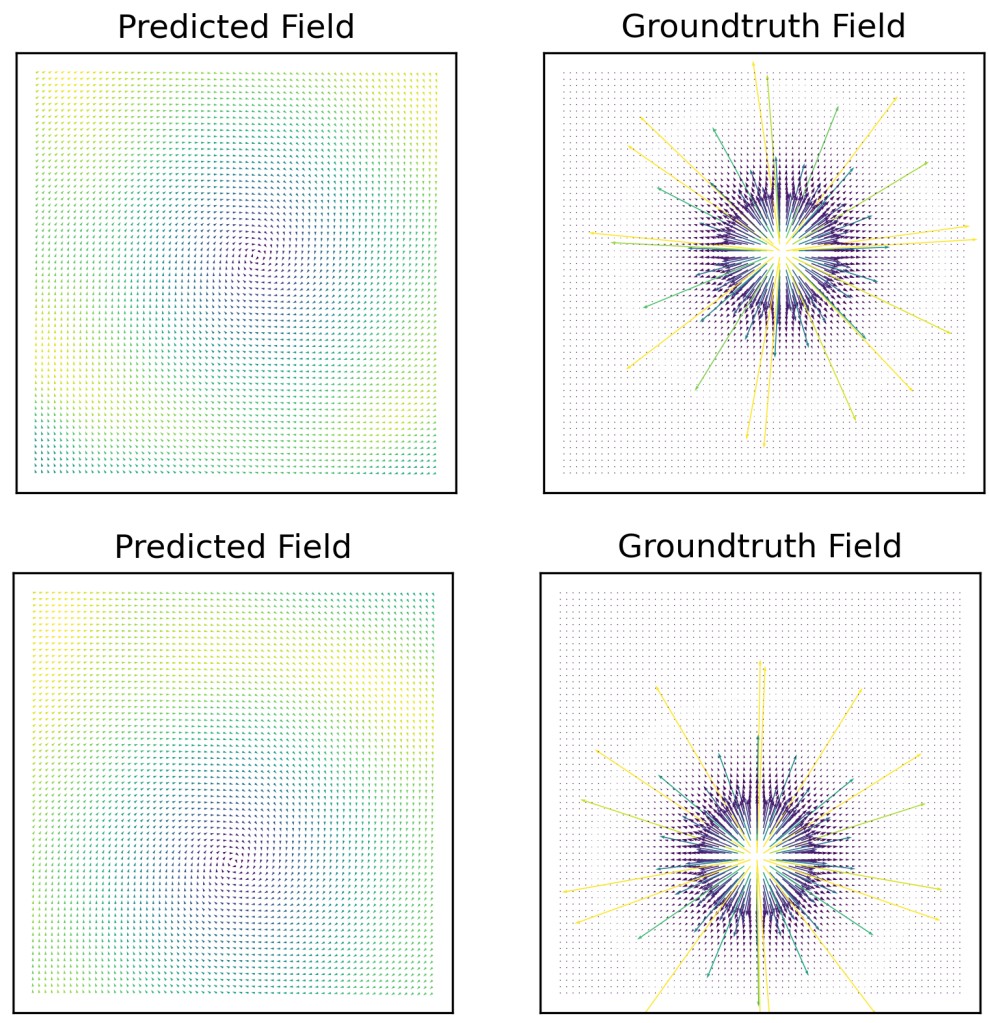

Figure 17: Learned *dynamic* fields (left) in 2D gravitational field setting vs groundtruth (right).

# E    Quantitative results

In all settings, we report the *total errors*, *i.e.* the mean squared errors of positions and velocities over time, $E(t) = \frac{1}{ND} \sum_{n=1}^{N} \|\mathbf{x}_n^t - \hat{\mathbf{x}}_n^t\|_2^2$. Following Kofinas et al. [24], we also separately report the $L_2$ norm *position errors*, $E_p(t) = \frac{1}{N} \sum_{n=1}^{N} \|\mathbf{p}_n^t - \hat{\mathbf{p}}_n^t\|_2$, and *velocity errors*, $E_u(t) = \frac{1}{N} \sum_{n=1}^{N} \|\mathbf{u}_n^t - \hat{\mathbf{u}}_n^t\|_2$.

## E.1    Electrostatic field

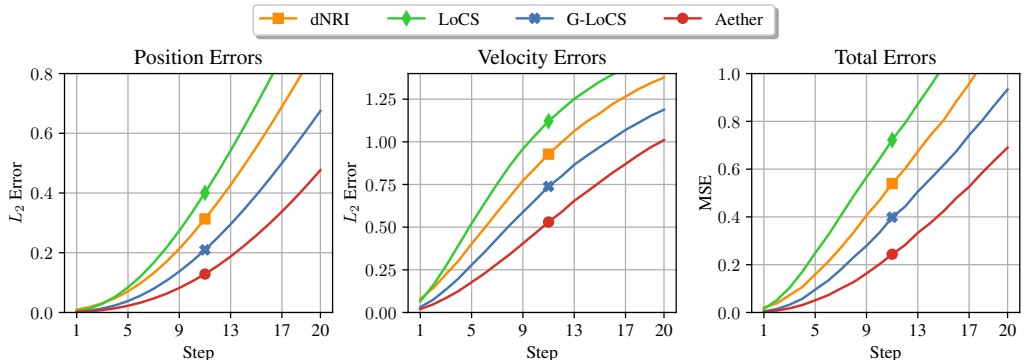

Figure 18: Results in the electrostatic field setting.

## E.2    InD

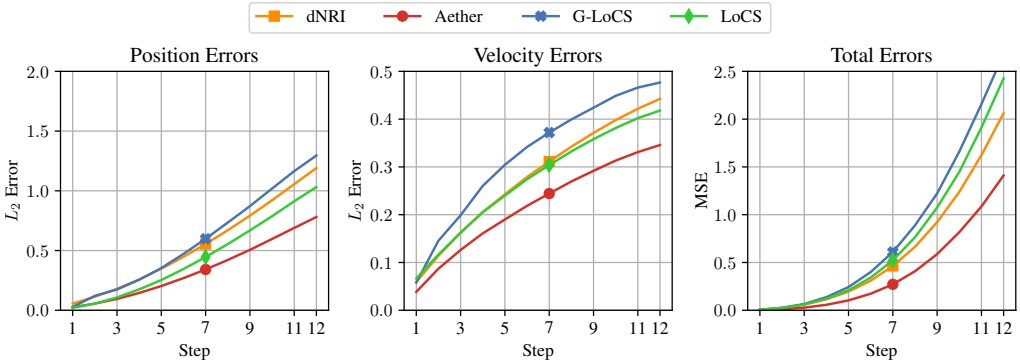

Figure 19: Results in inD.

## E.3 Gravity

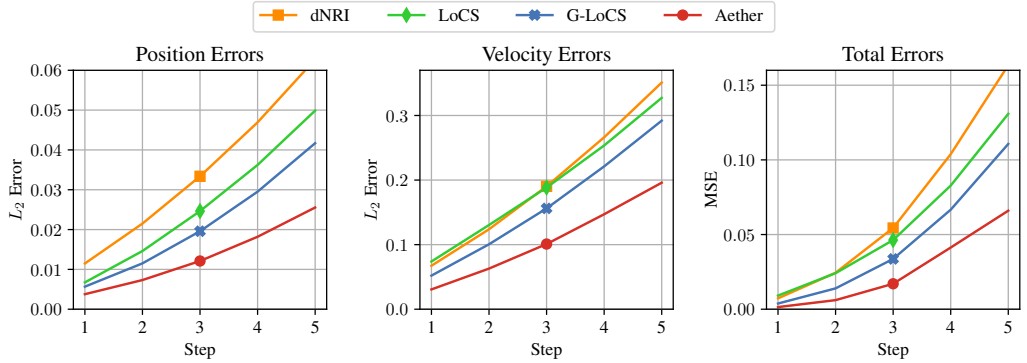

Figure 20: Results in the dynamic gravitational field setting.

## E.4 2D gravity

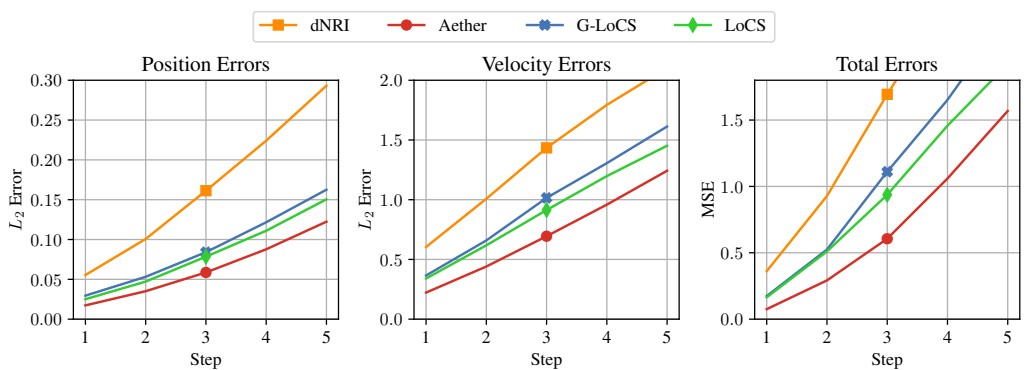

Figure 21: Results in the dynamic 2D gravitational field setting.

## E.5 Significance of the discovered field

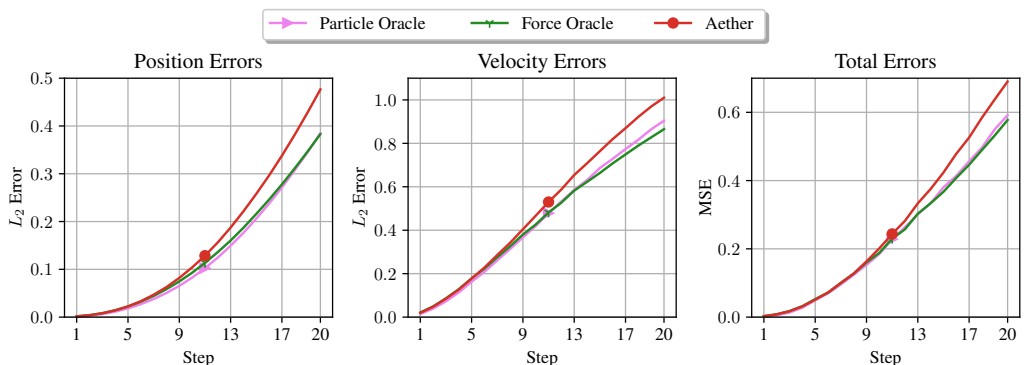

Figure 22: Ablation study on the significance of the discovered field. Results in the electrostatic field setting.

