# OpenReview forum: "Latent Field Discovery in Interacting Dynamical Systems with Neural Fields"
_NeurIPS.cc/2023/Conference — NeurIPS 2023 poster_

### Official Review · Reviewer_HXpt · 2023-07-03

**Soundness:** 2 fair
**Presentation:** 3 good
**Contribution:** 2 fair
**Rating:** 6
**Confidence:** 4

**Summary:**

In this paper, the authors address the often-overlooked influence of underlying field effects on the evolution of interacting systems, postulating the existence of latent force fields. They suggest utilizing neural fields for learning these dynamics. Equivariant networks, although commonly used, fail to capture the global information, a crucial shortcoming the authors aim to rectify. They propose to separate local object interactions (SE(3) equivariant and dependent on relative states) from the global field effects (dependent on absolute states).

Through the use of equivariant graph networks to model these interactions and their integration with neural fields, the authors construct a novel graph network to incorporate field forces. The authors' experimental results demonstrate their ability to successfully identify underlying fields within various scenarios, including charged particle settings, traffic scenes, and gravitational n-body problems. Furthermore, these findings prove the method's effectiveness in system learning and trajectory forecasting.

**Strengths:**

The structure and presentation of the paper are appropriately executed. The authors have used clear and precise language, which aids in the understanding of the complex concepts discussed. This ease of readability can prove beneficial for readers across various levels of familiarity with the subject matter, contributing to the paper's usefulness.

The concept of entangled equivariance, as introduced in this work, is a novel approach in system dynamics. This method, which focuses on factorizing out the influence of global fields from local interactions, provides a fresh insight into the understanding of interacting systems.

I really enjoyed the range of experiments conducted in the paper, involving diverse systems of interacting entities at different scales. The application of the proposed methodology across different contexts, such as charged particles, traffic scenes, and gravitational n-body problems, illustrates the method's broad applicability.

**Weaknesses:**

I found it interesting that there's no  explicit mention of the training objective. This omission can potentially lead to ambiguity in understanding the complete process and methodology involved. For a comprehensive study, it is crucial to explicitly specify each component, including the training objective, to ensure a thorough understanding of the research.

The training objective appears to be a reconstruction term (given Fig. 3), which may prove problematic in non-deterministic like trajectory forecasting. As it currently stands, the proposed approach will always predict the same future trajectory given the same input trajectory, a characteristic that is incongruous with the inherent non-deterministic nature of forward predictions in time, especially when the complete dynamical system is not fully known and explicitly defined. In addition, the approach is evaluated primarily in the context of future trajectory prediction, which only accentuates these concerns. I

Lastly, there seems to be an omission in the citations pertaining to probabilistic models of neural fields. Specifically, works such as Functa [1] and Diffusion Probabilistic Fields [2] are not referenced. Including these references would enhance the completeness of the literature review and underscore the authors' comprehensive understanding of the existing body of research.

[1] Dupont, Emilien, et al. "From data to functa: Your data point is a function and you can treat it like one." arXiv preprint arXiv:2201.12204 (2022).

[2] Zhuang, Peiye, et al. "Diffusion probabilistic fields." The Eleventh International Conference on Learning Representations. 2022.

**Questions:**

There are a few open question which are key to the formulation of the training objective of the proposed approach. What are the reasons for choosing a deterministic training objective (eg. L2 reconstruction) for a problem that is inherently non-deterministic. Wouldn't a model that predicts a distribution over future trajectories be conceptually preferred to model that predicts a single point estimate? I am happy to update my score after discussing these questions with authors.

**Limitations:**

Limitations and societal impact of the proposed approach is properly discussed

---

> ### Author Rebuttal · Authors · 2023-08-09
>
> We would like to thank the reviewer for their insightful comments. We appreciate that they find our method novel and insightful with broad applicability. Please find our answers below.
>
> ```
> I found it interesting that there's no explicit mention of the training objective. This omission can potentially lead to ambiguity in understanding the complete process and methodology involved. For a comprehensive study, it is crucial to explicitly specify each component, including the training objective, to ensure a thorough understanding of the research.
> ```
>
> Thank you for the recommendation. We discuss the training objective in detail in appendix A.1.2, lines 612-620. However, we acknowledge the importance of explicitly specifying each component, and we will include details about the training objective in the main document.
>
> ```
> What are the reasons for choosing a deterministic training objective (eg. L2 reconstruction) for a problem that is inherently non-deterministic. Wouldn't a model that predicts a distribution over future trajectories be conceptually preferred to model that predicts a single point estimate?
> ```
>
> In most experimental settings (except the Lorentz force field setting), our method is formulated as a VAE with latent edge types. The training objective is minimizing the negative ELBO, which comprises a reconstruction term (Gaussian log likelihood) and a KL divergence regularization term. As such, our method can predict multiple different trajectories by sampling different latent variables, or sampling from the output Gaussian distribution, instead of using just the mean as a point estimate.
>
> In the Lorentz force field setting, we indeed train only using the reconstruction term (MSE). While we agree that a probabilistic approach is more suitable to the task of future forecasting, we note, that this is in line with the majority of state-of-the-art works in the field, see [3, 4] as examples, as well as [5] that introduced this dataset.
>
> ```
> Lastly, there seems to be an omission in the citations pertaining to probabilistic models of neural fields. Specifically, works such as Functa [1] and Diffusion Probabilistic Fields [2] are not referenced. Including these references would enhance the completeness of the literature review and underscore the authors' comprehensive understanding of the existing body of research.
> ```
>
> Thank you for the suggestions. We agree that the line of works that includes Functa [1] and DPF [2] are related to our work, and we will gladly include these references in the related work for the camera ready version.
>
> ### References
>
> [1] Dupont, Emilien, et al. "From data to functa: Your data point is a function and you can treat it like one." ICML, 2022.
>
> [2] Zhuang, Peiye, et al. "Diffusion probabilistic fields". ICLR, 2022.
>
> [3] Satorras, Vıctor Garcia, et al. "E(n) equivariant graph neural networks". ICML, 2021.
>
> [4] Du, Weitao, et al. "SE(3) equivariant graph neural networks with complete local frames". ICML, 2022.
>
> [5] Brandstetter, Johannes, et al. "Geometric and physical quantities improve E(3) equivariant message passing". ICLR, 2022.

---

> > ### Comment · Reviewer_HXpt · 2023-08-15
> >
> > I have read the rebuttal and I am satisfied with the answers provided by the authors.

---

### Official Review · Reviewer_DhsC · 2023-07-05

**Soundness:** 3 good
**Presentation:** 3 good
**Contribution:** 3 good
**Rating:** 6
**Confidence:** 5

**Summary:**

In this paper, the authors have introduced a novel method, Aether, for discovering the latent field within dynamical systems in order to better model their dynamical evolutions. This is achieved by disentangle the interactions between objects and the effect of the external global field, conditioned on the states for the static field case or an extra latent state vector for the dynamic field case. The experiments on charged particles with external fields, Lorentz force field, traffic scenes, and gravitational field demostrate that Aether can learn to discover the latent external field and leverage the field information to better model the dynamics of the entire system.

**Strengths:**

1. The paper identifies an important problem of latent field discovery which is of good originality.

2. The presentation of the paper is clear and easy to read. The method is well-designed and easy to follow.

3. The experiments are carefully designed which cover a wide range of datasets including physical systems and even traffic scences data.

**Weaknesses:**

1. The experiments only comprise relatively simple scenarios and even methods compared against (see Q1 and Q2). It would be better to showcase the effectiveness and impact of the method on more challenging and complex datasets/benchmarks.

2. Some experiment results seem confusing and might require further explanations (see Q3).

**Questions:**

Q1. Most of the datasets used in the paper are artificial/simulation data. Are there any practical application of the field discovery method on more real-world data (besides the traffic data)? Adding more real-world datasets will largely enhance the credibility of the proposed approach.

Q2. Even for the traffic data, it would be of great significance to compare the method with state-of-the-art traffic prediction approaches, while in this paper only weak baselines are considered for this dataset. It is unclear that to what extent the current method would impact these fields of practical applications.

Q3. Results in Figure 4 may require additional justifications. The reported MSEs of all methods seem to be around 0.6 at timestep 20 on charged NRI dataset, while in the original paper [1] the numbers were around the scale of 1e-3 when timestep = 20. Are all models properly trained? Moreover, it would be helpful to include ground truth trajectories in the visualizations to see whether the models can offer visually reasonable predictions, as per previous works.

[1] Kipf et al. Neural Relational Inference for Interacting Systems. 2018.


**Limitations:**


The authors have sufficiently discuss the limitations of their method.

---

> ### Author Rebuttal · Authors · 2023-08-09
>
> We would like to thank the reviewer for their insightful comments. We appreciate that they find our problem statement original and our method well-designed. Please find our answers below.
>
> ```
> Q1. Most of the datasets used in the paper are artificial/simulation data. Are there any practical application of the field discovery method on more real-world data (besides the traffic data)? Adding more real-world datasets will largely enhance the credibility of the proposed approach.
> ```
>
> While fields underlie many real-world systems, to the best of our knowledge, there are no established real-world datasets that include fields. As the community moves beyond strict equivariance, we expect that more and more real-world datasets that incorporate fields will be established.
>
> ```
> Q2. Even for the traffic data, it would be of great significance to compare the method with state-of-the-art traffic prediction approaches, while in this paper only weak baselines are considered for this dataset. It is unclear that to what extent the current method would impact these fields of practical applications.
> ```
>
> Thank you for your suggestion. We perform a new experiment, where we integrate our method in EqMotion [5], a recent equivariant method with state-of-the-art performance on trajectory forecasting. We incorporate Aether in EqMotion by treating the predicted forces as geometric features similar to velocities. Namely, after computing the forces for each object at each timestep, we compute the magnitudes of the force vectors and the force angle sequence, i.e. the angles between forces in consecutive timesteps. We concatenate these quantities to the existing features for the feature initialization step.
>
> We train and evaluate EqMotion and Aether+EqMotion on inD following our experimental setup. Here we report the final total error, i.e. the MSE at timestep 12. EqMotion has an MSE of 1.104, while Aether+EqMotion has an MSE of 0.962. We see that Aether is beneficial even for a state-of-the-art trajectory forecasting method, which further strengthens our claims.
>
> ```
> Q3. Results in Figure 4 may require additional justifications. The reported MSEs of all methods seem to be around 0.6 at timestep 20 on charged NRI dataset, while in the original paper [1] the numbers were around the scale of 1e-3 when timestep = 20. Are all models properly trained? Moreover, it would be helpful to include ground truth trajectories in the visualizations to see whether the models can offer visually reasonable predictions, as per previous works.
> ```
>
> There are three very important differences between our electrostatic field experiment and the original charged particles dataset [1]. First, these are two different datasets; our electrostatic field dataset extends the original dataset by adding an electrostatic field. Trajectory forecasting in the electrostatic field is a much harder task, since the unobserved field can drastically impact future trajectories. Second, the y-axis between the two figures is actually different; both measure the MSE, but NRI reports the normalized error, i.e. the error between min-max normalized groundtruth and predictions, while we report the unnormalized error following [4], i.e. the error in the original data scale, to ensure fair comparison across different normalization functions. Finally, as we mention in appendix B.1, following [2, 3], we have removed virtual borders that cause elastic collisions.
>
> Please see appendix D.1 for visualizations and comparison with groundtruth future trajectories, as well as videos in the electrostatic_predictions directory in the supplementary material. They demonstrate that our method make reasonable predictions.
>
> ### References
>
> [1] Kipf et al. “Neural Relational Inference for Interacting Systems”. ICML, 2018.
>
> [2] Satorras, Vıctor Garcia, et al. "E(n) equivariant graph neural networks". ICML, 2021.
>
> [3] Fuchs, Fabian, et al. "SE(3)-transformers: 3d roto-translation equivariant attention networks." NeurIPS, 2020.
>
> [4] Kofinas, Miltiadis, et al. "Roto-translated local coordinate frames for interacting dynamical systems." NeurIPS, 2021.
>
> [5] Xu, Chenxin, et al. "EqMotion: Equivariant Multi-agent Motion Prediction with Invariant Interaction Reasoning". CVPR, 2023.

---

> > ### Comment · Reviewer_DhsC · 2023-08-12
> > **Thank you!**
> >
> > Thank you for the efforts you have made in addressing my concerns. I have no futher question and hope the authors will incorporate the new results and explanations into the final paper. I have increased the score.

---

### Official Review · Reviewer_8BnH · 2023-07-07

**Soundness:** 3 good
**Presentation:** 3 good
**Contribution:** 3 good
**Rating:** 6
**Confidence:** 3

**Summary:**

The main assumption of this paper is that local object interactions generally happen within external global fields. Based on this assumption, the paper then proposes to disentangle the local from the global interactions, respectively, by learning both an equivariant graph network, capturing the local interactions, and a neural network, capturing the global interactions. The latter network modifies the behavior of the graph network. The experimental results of the paper demonstrate that one can accurately discover the underlying fields in charged particles settings, traffic scenes, and gravitational n-body problems, and effectively use them to learn the system and forecast future trajectories.

**Strengths:**

The idea of considering local interactions happening in the presence of global fields is an interesting one, and seems to be new. So is the approach, learning the external fields, and the experimental validation demonstrate an increased accuracy of objects-trajectory prediction.

**Weaknesses:**

The results of this work seem to be confined to physics applications.

**Questions:**

Is it possible to generalize the ideas in this paper to a structuring mechanism relevant to a more general class of ML applications?

**Limitations:**

This work only considers fields that do not react to the observable environment.

---

> ### Author Rebuttal · Authors · 2023-08-09
>
> We would like to thank the reviewer for acknowledging the novelty of our idea and approach. Please find our answers below.
> ```
> Is it possible to generalize the ideas in this paper to a structuring mechanism relevant to a more general class of ML applications?
> ```
> The notion of the field is tightly linked with a global phenomenon that occurs in the continuous (Euclidean) space. This space could be underlying a geometric structure such as a geometric graph or an image grid, or be a continuous relaxation of such a structure.
> We can draw a lot of qualitative similarities with works on graph networks that explore virtual global nodes. For example, the work of Battaglia et al. [2] explored graphs with global attributes. While they can only model constant global attributes, our fields go a step further, since we now have a global function that can take different values based on the node states. In other words, global attributes can be seen as a special case of our method, one where our predicted field is constant everywhere.
> Furthermore, in [2], global properties of the graph can be derived from global attributes. Similarly, we expect to be able to derive graph properties from operating on the global neural field, by using a method that operates on neural fields, e.g. the recently popularized Functa [1].
>
> We believe that this approach could be used to approximate and replace graph sub-structures or image grids, and we are excited to see the community explore this research direction.
>
> ### References
>
> [1] Dupont, Emilien, et al. "From data to functa: Your data point is a function and you can treat it like one." ICML, 2022.
>
> [2] Battaglia, Peter W., et al. "Relational inductive biases, deep learning, and graph networks". 2018.

---

> > ### Comment · Reviewer_8BnH · 2023-08-18
> > **Answer to rebuttal**
> >
> > Thank you very much for your answers. I am happy with your response.

---

### Official Review · Reviewer_hUPP · 2023-07-14

**Soundness:** 3 good
**Presentation:** 2 fair
**Contribution:** 3 good
**Rating:** 6
**Confidence:** 4

**Summary:**

This paper proposes to learn the latent fields governing the dynamics of interacting objects in a system without observations of the field. The authors propose unconditional and conditional latent field models to learn the static and dynamic fields respectively. The learned latent field signals are represented in local coordinate representations and treated as additional node features in graph learning to predict future node trajectories. Specifically, the paper adopts a recent VAE framework to make predictions in an autoregressive way.

**Strengths:**

* The general idea of discovering latent neural fields is very interesting and plays an important role in modeling many real-world problems and scientific tasks.
* The proposed idea looks simple yet effective in learning the latent fields. It is flexible and potentially can be applied to many graph learning frameworks.
* The quantitative and qualitative results showcase the proposed idea and clearly show the performance gains compared with the previous works.

**Weaknesses:**

Unclear notations:
* The shape of different variables are often not explained clearly. It’s not easy to infer from the context and therefore not self-contained.
* Eq (1): how to understand the shape of angular position $\boldsymbol{Q(w_i^t)}$ and $\boldsymbol{\tilde{R}(w_i^t)}$? The notations do not seem to be self-contained.
* Eq (2) - (3) what are the definitions of $\boldsymbol{h_{j,i}^t}$?
* Definition of angle in L169 uses the symbol $\boldsymbol{\omega}$, is it the same as in L74?
* L94 uses the symbol $\boldsymbol{h}$, is it related to h in Eq (2)?
* Latent code $\boldsymbol{z}$ and the latent graph edge type $\boldsymbol{z}$ in the appendix are using the same letter, which is very confusing.

L44 and below: In the example of the N-body system, if the E(3) symmetry holds, then SE(3) symmetry must hold since SE(3) is a special case of E(3) without reflection. To the reviewer, this example doesn’t explain the pitched point that SE(3) equivariance doesn’t always hold.

About the combination of latent field learning and VAE:
* The proposed models use latent edge-based VAE to predict the future node states. It’d be better to contain the VAE formulation in the main paper briefly to be self-contained.
* The idea of learning latent fields can be applied in non-VAE frameworks for trajectory predictions as well. It’d be more convincing to show these experimental results.

About neural field modeling.
* How important is the assumption of static and dynamic fields for learning? Can the model learn different fields in a data-driven manner?
* To the reviewer, either conditional or unconditional neural fields modeling methods can be used to learn both static or dynamic fields. It’s not convincing without ablation experiments to justify the model design w.r.t. types of fields.
* L155-L162: the naming of unconditional and conditional neural fields look confusing to the reviewer, as both are functions of input states essentially. Although they are different in architecture design, both are neural functions of the input trajectories.
* Figure 3 shows the conditional neural field setup for dynamic fields, while most of the experiments are using static fields. To the reviewer, this is a bit misleading and does not showcase the proposed model properly. Please add both static and dynamic fields setup in the figure.

About G-LoCS baseline:
* How is the artificial velocity defined in the experiments? The details are vague to the reviewer.
* This method is essentially an additional baseline with node feature augmentation and is not a novel contribution to the reviewer.

**Questions:**

Please see comments above.

---

> ### Author Rebuttal · Authors · 2023-08-10
>
> We would like to thank the reviewer for their insightful comments. We appreciate that they find the idea of field discovery very interesting, and our method simple and effective. Please find our answers below.
>
> ```
> Unclear Notation
> ```
> - We describe most shapes in detail in appendix A. We will incorporate the variable sizes in our equations for the updated version of the manuscript.
> - The matrix $\mathbf{Q}$ constitutes the matrix representation of the angular position $\boldsymbol{\omega}$, i.e. it is a rotation matrix in SO(d), d=2, 3. Hence, depending on the dimensionality of the problem, it is a $2 \times 2$ or $3 \times 3$ matrix. The matrix $\mathbf{\tilde{R}}$ is the direct sum of three rotation matrices $\mathbf{Q}$, i.e. it is a block diagonal matrix where each block is a matrix $\mathbf{Q}$. Its dimensions are $6 \times 6$ in 2 dimensions or $9 \times 9$ in 3 dimensions.
> - $\mathbf{h}^t_{j,i}$ are the intermediate edge features (from node $j$ to node $i$) or “messages” in the message passing network nomenclature. The superscript $t$ denotes the actual timestep.
> - Yes, the definition of angle or angular position $\boldsymbol{\omega}$ in lines 74 and 169 is the same. In line 169, we define $\boldsymbol{\omega}$ for the more general case of 3 dimensions; in 2 dimensions we only have one angle and $$\boldsymbol{\omega}$  = \theta$.
> - Indeed, we use the same letter $\mathbf{z}$ to describe two different variables, the latent code and the latent graph edge type. We will fix the notation for the camera ready version of the paper.
>
> ```
> In the example of the N-body system, if the E(3) symmetry holds, then SE(3) symmetry must hold since SE(3) is a special case of E(3) without reflection. To the reviewer, this example doesn’t explain the pitched point that SE(3) equivariance doesn’t always hold.
> ```
>
> The point here is that the strict E(3) symmetry does not always hold, because it might be violated by external force fields, which are either unknown or not subject to transformation. As an example, an n-body system might evolve under the influence of an unobserved black hole that generates a gravitational field. Then, the observed dynamics would not be strictly equivariant any more, as we cannot transform the unobserved quantities.
>
> ```
> The proposed models use latent edge-based VAE to predict the future node states. It’d be better to contain the VAE formulation in the main paper briefly to be self-contained.
>
> The idea of learning latent fields can be applied in non-VAE frameworks for trajectory predictions as well. It’d be more convincing to show these experimental results.
> ```
>
> Thank you for the suggestion. The VAE formulation is orthogonal to our method; it serves as an implementation detail, and the method can also work without the VAE component. Case in point, our model in the Lorentz force field setting does not include a VAE (this also enables a fairer comparison with other baselines). See appendix A.1.3 for the non-VAE implementation of our method. For the camera ready version, we will clarify that the VAE is orthogonal to our method, and highlight that non-VAE is also possible.
>
>
> ```
> To the reviewer, either conditional or unconditional neural fields modeling methods can be used to learn both static or dynamic fields. It’s not convincing without ablation experiments to justify the model design w.r.t. types of fields.
> ```
>
> Unconditional neural fields cannot be used to learn dynamic fields, since they would only learn an average force field across the dataset. On the other hand, a conditional neural field could be used to learn a static field. In that case, the neural field should learn to ignore the latent vector $\mathbf{z}$, since the generated field should be identical regardless of the input system; we expect its performance to match the unconditional field. We verify this hypothesis with an ablation study on the Lorentz force field, where we train and evaluate our method using a conditional field. The MSE is now 0.0131, while the original unconditional method had an MSE 0.0129.
>
> ```
> L155-L162: the naming of unconditional and conditional neural fields look confusing to the reviewer, as both are functions of input states essentially.
> ```
>
> We borrow the terminology of conditional and unconditional neural fields from [1]. Unconditional neural fields are not functions of the input states, on the contrary, they are independent of the input states. We can sample the neural field in arbitrary positions; in practice we sample it at the input states because we have supervision there. On the other hand, the conditional neural field is a function of the input states, since they are used to generate the latent vector $\mathbf{z}$ used to condition the neural field. We will clarify terminology in the introduction.
>
> ```
> About G-LoCS baseline, how is the artificial velocity defined in the experiments?
> ```
>
> As we mention in the background section of the paper, LoCS [2] uses velocities as a proxy to estimate angular positions. G-LoCS introduces an auxiliary node that corresponds to the global coordinate frame. We define its velocity to match the x-axis, e.g. $\mathbf{u} = (1,0)^T$ in 2 dimensions, such that we can compute a non-degenerate angular position (frame).
>
> ```
> The G-LoCS baseline is essentially an additional baseline with node feature augmentation and is not a novel contribution to the reviewer.
> ```
>
> G-LoCS is an approximately equivariant graph network that retains the strengths of equivariance through local coordinate frames, but it can also capture global phenomena through the global coordinate frame. We consider it to be a conceptually simple yet novel approximately equivariant graph network with minimum computational overhead, as well as an additional baseline.
>
> ### References
>
> [1] Xie, Yiheng, et al. "Neural fields in visual computing and beyond." 2022.
>
> [2] Kofinas, Miltiadis, et al. "Roto-translated local coordinate frames for interacting dynamical systems." NeurIPS, 2021.

---

> > ### Comment · Reviewer_hUPP · 2023-08-14
> >
> > Thanks for your feedback.
> >
> > As a minor suggestion, I recommend rephrasing the original text concerning `strict E(3) symmetry` to enhance its clarity (for instance, by employing the black hole example you provided in the context of external force fields).
> > > N-body systems from physics, for example, exhibit E(3) symmetries, since gravitational forces only depend on relative positions. Dynamics, however, may be influenced by external force fields, which are either unknown or not subject to transformations.
> >
> > Regarding the unconditional and conditional neural fields, the original paper and the reviewer's rebuttal appear somewhat unclear. The original text states in line 157:
> > >  we use an **unconditional neural field**, i.e. a neural field that is **a function only of the input states**, as the field values are common across data sample.
> >
> > The authors explain in the rebuttal with the following statement:
> > > **Unconditional neural fields are not functions of the input states**, on the contrary, they are independent of the input states.
> >
> > These two statements seem to contradict each other.
> >
> > Furthermore, the original paper mentions:
> > > In contrast, for dynamic fields, we use a **conditional neural field**, i.e. a neural field that also depends on a latent vector $z  \\in \\mathbb{R}^{D\_z}$ that represents the underlying field. The **latent z will be inferred from the input trajectories** and can be thought of as representing unusual non-equivariant dynamics.
> >
> > What distinguishes the `input states` of the unconditional field from the `input trajectories` of the conditional field? Are they perceived as the same from the network input perspective?
> >
> > Lastly, this question appears to be disregarded in the rebuttal:
> > > How important is the assumption of static and dynamic fields for learning? Can the model learn different fields in a data-driven manner?
> >
> > As the authors acknowledge in the rebuttal, **`a conditional neural field could be used to learn a static field`**. Can we consistently utilize conditional field models to learn both types of data and eliminate the need for assuming the field type beforehand? Regarding empirical performance, how crucial is this assumption across a broader range of experiments?
> >
> > Specifically, the authors mention that for the Lorentz force field (static field) experiment, the MSE of the conditional model is 0.0131, while the original unconditional method had an MSE of 0.0129. This performance difference of **0.0002** seems negligible to the reviewer. Then, what is the rationale behind the unconditional model?
> >
> > Thanks.

---

> > > ### Author Response · Authors · 2023-08-17
> > > **Rebuttal by Authors**
> > >
> > > We would like to thank the reviewer for their detailed comments. Please find our answers below.
> > >
> > > ```
> > > As a minor suggestion, I recommend rephrasing the original text concerning strict E(3) symmetry to enhance its clarity (for instance, by employing the black hole example you provided in the context of external force fields).
> > > ```
> > >
> > > Thank you for the suggestion. We will rephrase the original text to enhance its clarity regarding E(3) and SE(3) symmetries.
> > >
> > > ```
> > > The statements “we use an unconditional neural field, i.e. a neural field that is a function only of the input states, as the field values are common across data sample” and “Unconditional neural fields are not functions of the input states, on the contrary, they are independent of the input states” seem to contradict each other.
> > > ```
> > >
> > > We apologize for the confusion. The two notions of functions come from different perspectives, namely a practical perspective and a theoretical/mathematical perspective. From a mathematical perspective, the field is a function of x-y coordinates, x-y constituents of a velocity vector, etc. One could compute the field at any valid tuple of positions, velocities etc. In that sense, the field is independent of the input states. In other words, the unconditional neural field will make the same predictions regardless of the inputs.
> > >
> > > In practice, during training, we only sample the field at the positions/velocities that coincide with the states of the input objects, since we only have supervision there. As such, with a slight abuse of notation, we write $\mathbf{f}_i = f(\mathbf{v}_i)$, where $\mathbf{v}_i$ is the state of node $i$.
> > >
> > > We acknowledge that this terminology mixup can confuse the reader. We will incorporate parts of this discussion in the camera ready version to increase clarity. Namely, we will mention that “unconditional neural fields are not functions of the input states”, and remove the term “input states” in other occurrences concerning the neural field to avoid confusion.
> > >
> > > ```
> > > What distinguishes the input states of the unconditional field from the input trajectories of the conditional field? Are they perceived as the same from the network input perspective?
> > > ```
> > >
> > > The terms “input states” and “input trajectories” describe very similar concepts, yet they are distinct. The input states denote the information about nodes-objects (positions, velocities, orientations) for a single timestep. As we mention in lines 129-130 in the original manuscript, these states - or a subset of the state variables (e.g. positions only) - are used as input to the neural field. The term input trajectories denotes the whole system we have access to, including the temporal information, i.e. the input states for a number of timesteps.
> > >
> > > In the conditional field case, input trajectories are used to create the latent vector $\mathbf{z}$. The latent vector is used alongside sampled positions/velocities to compute the field. This means that the conditional field depends on the (spatiotemporal) inputs, and would produce different predictions for different inputs.
> > >
> > > ```
> > > How important is the assumption of static and dynamic fields for learning? Can the model learn different fields in a data-driven manner?
> > > ```
> > >
> > > We apologize for the omission. Our method discovers dynamic fields conditioned on the observable input system. As such, the working hypothesis is that the input states cover the domain of the field we are trying to discover. If we do not have enough coverage, then we cannot discover the field effectively. Similarly, we cannot model fields that react to the system, unless we have enough spatial coverage for each temporal snapshot of the field.
> > >
> > > ```
> > > As the authors acknowledge in the rebuttal, a conditional neural field could be used to learn a static field. Can we consistently utilize conditional field models to learn both types of data and eliminate the need for assuming the field type beforehand? Regarding empirical performance, how crucial is this assumption across a broader range of experiments?
> > > Specifically, the authors mention that for the Lorentz force field (static field) experiment, the MSE of the conditional model is 0.0131, while the original unconditional method had an MSE of 0.0129. This performance difference of 0.0002 seems negligible to the reviewer. Then, what is the rationale behind the unconditional model?
> > > ```
> > >
> > > While a conditional neural field be used to learn a static field, this can come at the cost of increased training and inference time, as well as redundant computational resources and model parameters. Indeed, the performance difference of 2e-4 is rather negligible. However, we would argue that when there is expert knowledge that the field at hand is a static field, then the unconditional neural field is the preferred choice. In the absence of such knowledge, e.g. on an exploratory analysis for underlying fields, then the conditional neural field would be preferable.

---

> > > > ### Comment · Reviewer_hUPP · 2023-08-18
> > > >
> > > > I appreciate your responses and most of my concerns have been addressed. My curiosity lies in understanding the practical consequences of the described effects, such as longer training and inference times, along with duplicated computational resources and model parameters, stemming from using conditional neural fields for static fields. Is it feasible to **quantitatively** demonstrate the concrete overhead in terms of training and inference time, model size, and computational demands among the experiments detailed in the paper? Providing such comparisons would better validate your claim within the context of practical machine learning applications.

---

> > > > > ### Author Response · Authors · 2023-08-19
> > > > >
> > > > > We would like to thank the reviewer for their detailed comments. Please find our answers below.
> > > > >
> > > > > ```
> > > > > Is it feasible to quantitatively demonstrate the concrete overhead in terms of training and inference time, model size, and computational demands among the experiments detailed in the paper?
> > > > > ```
> > > > >
> > > > > We would like to thank the reviewer for their suggestion. In the following table, we report the training time per minibatch, the inference time, and the number of parameters for each model in the Lorentz force field setting. While the (unconditional) Aether has 2,515 more parameters than the LoCS backbone, the conditional Aether has 9,985 more parameters on top of the regular Aether, as well as 27% higher inference time. We will include these quantitative results in the camera ready version of the paper.
> > > > >
> > > > > Method | Params | Inference Time | Training time / minibatch
> > > > > ---: | ---: | ---: | ---: |
> > > > > EGNN | 134,020 | 0.0028 | 0.0037 |
> > > > > LoCS | 130,307 | 0.0033 | 0.0038 |
> > > > > Aether  | 132,822 | 0.0037 | 0.0049 |
> > > > > Conditional Aether | 142,807 | 0.0047 | 0.0053

---

> > > > > > ### Comment · Reviewer_hUPP · 2023-08-19
> > > > > >
> > > > > > Thanks for your feedback. I have no more concerns about the paper and I have raised my score.

---

### Author Rebuttal · Authors · 2023-08-10

### Reponse to question by Reviewer hUPP
```
Figure 3 shows the conditional neural field setup for dynamic fields, while most of the experiments are using static fields. To the reviewer, this is a bit misleading and does not showcase the proposed model properly. Please add both static and dynamic fields setup in the figure.
```

Thank you for the suggestion. We acknowledge that this can be misleading, and we will update the figure for the camera ready version of the paper. The architecture for the static field settings is almost the same, with the exception of the graph aggregation module that generates the latent vector $z$ and the FiLM layers that modulate the neural field.
Please see the update figure attached in the pdf.

---

### Decision · Program_Chairs · 2023-09-21

**Decision:**

Accept (poster)

**Comment:**

The paper extends previous works on learning latent interactions for observations of the dynamics of interacting objects to include learning latent external fields. This is an important extension to the existing literature and the paper shows applications in both physical systems as well as traffic scenes. The authors use a novel architecture to address this problem that takes into account the appropriate symmetries. As such we decided that paper should be accepted to publication. However, there were many remarks on the writing/notations and we hope the writters will take the time to incoporate these remarks into the final version.